# Tumor microenvironment governs the prognostic landscape of immunotherapy for head and neck squamous cell carcinoma: A computational model-guided analysis

Priyan Bhattacharya[1], Alban Linnenbach[2], Andrew P. South[3],
Ubaldo Martinez-Outschoorn[4], Joseph M. Curry[2], Jennifer M. Johnson[2,4],
Larry A. Harshyne[5], Mỹ G. Mahoney[3], Adam J. Luginbuhl[2], Rajanikanth Vadigepalli[1]*

1 Department of Pathology and Genomic Medicine, Thomas Jefferson University, Philadelphia, Pennsylvania, United States of America, 2 Department of Otolaryngology, Head and Neck Surgery, Thomas Jefferson University, Philadelphia, Pennsylvania, United States of America, 3 Department of Pharmacology, Physiology, and Cancer Biology, Thomas Jefferson University, Philadelphia, Pennsylvania, United States of America, 4 Department of Medical Oncology, Thomas Jefferson University, Philadelphia, Pennsylvania, United States of America, 5 Department of Microbiology and Immunology, Thomas Jefferson University, Philadelphia, Pennsylvania, United States of America

* Rajanikanth.Vadigepalli@jefferson.edu

## Abstract

Immune checkpoint inhibition (ICI) has emerged as a critical treatment strategy for squamous cell carcinoma of the head and neck (HNSCC) that halts the immune escape of the tumor cells. Increasing evidence suggests that the onset, progression, and lack of/no response of HNSCC to ICI are emergent properties arising from the interactions within the tumor microenvironment (TME). Deciphering how the diversity of cellular and molecular interactions leads to distinct HNSCC TME subtypes subsequently governing the ICI response remains largely unexplored. We developed a cellular-molecular model of the HNSCC TME that incorporates multiple cell types, cellular states, and transitions, and molecularly mediated paracrine interactions. Simulation across the selected parameter space of the HNSCC TME network shows that distinct mechanistic balances within the TME give rise to the five clinically observed TME subtypes such as immune/non-fibrotic, immune/fibrotic, fibrotic only and immune/fibrotic desert. We predict that the cancer-associated fibroblast, beyond a critical proliferation rate, drastically worsens the ICI response by hampering the accessibility of the CD8+ killer T cells to the tumor cells. Our analysis reveals that while an Interleukin-2 (IL-2) + ICI combination therapy may improve response in the immune desert scenario, Osteopontin (OPN) and Leukemia Inhibition Factor (LIF) knockout with ICI yields the best response in a fibro-dominated scenario. Further, we predict Interleukin-8 (IL-8), and lactate can serve as crucial biomarkers for ICI-resistant HNSCC phenotypes. Overall, we provide an integrated quantitative framework that explains a wide range of TME-mediated resistance mechanisms for

**Data availability statement:** The necessary MATLAB code and data required to reproduce the figures can be accessed at https://github.com/Daniel-Baugh-Institute/HNSCC_TME_Model

**Funding:** This project was supported by the Pennsylvania Commonwealth Universal Research Enhancement Program (CURE to AL). This project utilized the Biostatistics, Genomics, and Flow Cytometry Shared Resources at the Sidney Kimmel Cancer Center, supported by the National Cancer Institute Core grant (P30 CA056036 to AL). AS was funded by National Institute of Health (R01CA244522 to AS). The funders had no role in study design, data collection and analysis, decision to publish, or preparation of the manuscript.

**Competing interests:** The authors have declared that no competing interests exist.

HNSCC and predicts TME subtype-specific targets that can lead to an improved ICI outcome.

## Author summary

Increasing evidence suggests that the onset, progression, and lack of/no response of Head and Neck Squamous cell carcinoma (HNSCC) to immune checkpoint inhibitor (ICI) are emergent properties arising from the interactions within the tumor microenvironment (TME). Developing a mechanistic insights into how the compositional diversity of the TME determines the outcome of immunotherapy remains largely unexplored in the context of HNSCC. In this work, we developed a mathematical model that integrates the existing knowledge about the diverse forms of cytokines mediated cell-cell interactions (paracrine, autocrine, transition) within the TME to unpack (a) the mechanisms behind the emergence of different clinically observed TME subtypes such as fibrotic only, immune/fibrotic, non-fibrotic, and desert and (b) how these subtypes govern the response to immunotherapy. Subsequently, the model-guided approach enables us to propose potential biomarkers (IL-8, and lactate) for resistant phenotypes, and to identify subtype-specific target species (IL2 for fibrotic only, OPN and LIF knockout for a subclass of immune/fibrotic) for circumventing ICI resistance. Overall, the integrated quantitative framework explains a wide range of TME-mediated resistance mechanisms for HNSCC and predicts TME subtype-specific targets that can lead to an improved ICI outcome.

## Introduction

Squamous cell carcinoma of the head and neck (HNSCC) is a collection of head and neck cancers typically initiated from the mucosal cells in the larynx, pharynx, and oral cavity [1,2]. Recent clinical observations suggest that the response to immunotherapy strongly correlates with the cellular composition of the associated tumor microenvironment (TME) [3,4]. These observations necessitate elucidating the non-tumor cells' precise role in the survival and growth of tumor cells. The existing targeted therapies focusing on the intrinsic signaling factors within the tumor cells underplay the prospect of the TME functioning as a 'networked system' with emergent properties [5–7]. On the other hand, immunotherapies such as immune checkpoint inhibitors (ICI) attempt to strengthen the immune response of the TME through the elimination of specific molecular programs (such as programmed death-1 (PD1)/ programmed death ligand-1 (PDL1) or α-CTLA4) crucial for the immune escape of tumor cells [8]. Although the ICI treatments, under certain conditions, yield favorable outcomes, the majority of the HNSCC patients remain non-responsive to ICI treatment (only 13–31% of HNSCC patients respond to an Nivolumab/ Durvalumab-based therapy [9]). This indicates a need for the analysis of the interactions of different TME

components (tumor, immune, and fibrotic) towards a comprehensive understanding of the possible mechanisms for insensitivity to immunotherapy.

Mathematical modeling of the tumor cell population can provide a quantitative understanding of the role of TME constituents and estimate the patient-specific anti-cancer dosage amount [10–12]. Depending on the granularity, the existing mathematical models can be divided into at least three distinct categories— i) gene level, ii) cell population-based, and iii) cell-state transition networks. The gene-level modeling begins with the underlying gene regulatory network of the cell in consideration (tumor cells in this context) [13–15]. The multi-state nature of the dynamical system constituted from modeling the gene regulatory networks points to the existence of different cell states and their distinct responses to therapeutic strategies [13,16–20]. Modeling a gene regulatory network requires dealing with a large dimensional system with highly non-linear dynamics, making the model less intuitive and inconvenient for theoretical interventions. Therefore, although the gene-level analysis inspires novel targeted therapies focused on the tumor cells, a TME-wide understanding may not be feasible with this framework.

Cell-population-based networks overcome the dimensionality issue at gene levels by modeling the interaction between different cell types and the response of tumor cells to anti-tumor therapies [21–25]. The cell population-based models may miss out on capturing the inherent complexity in reliable detail if the functional heterogeneity for a given cell type is not taken into consideration. For instance, each cell type can have multiple distinct functional cell states with opposite effects on the growth of the tumor cells.

The classification of the cell states vis-a-vis the functionality within the tumor microenvironment have been a topic of sustained interest that includes a wide range of methods ranging from immunohistochemistry and immunofluorescence to clustering of the high-throughput single-cell transcriptome of the tumor [26–28]. The cellular states for each cell type can transition among themselves in the presence of specific biological promoters [26–30]. Further, two distinct cell states from different cell types can affect the proliferation of each other via either paracrine interactions or cytokine secretions [31–33].

The cell-state transition models, by design, can potentially circumvent the explainability issues encountered in cell population-based models. Further, since the number of cell states represents the cardinality of the attractor sets in the underlying regulatory dynamics, the model's dimensionality is reduced from the gene-level models [16]. Although several scholarly interventions have attempted to propose cell state models in the context of cancer, extension of the same to the entire TME for cancer (particularly HNSCC) remains an open area of study.

Considering the limitations and possibilities of different modeling paradigms, this work operates at the cell state level, wherein we reconstruct an essential interaction atlas for HNSCC based on an extensive literature survey. In the state space form, we represent the population and concentration of the cellular states and molecular species. Via an exhaustive scanning across the model parameter space, we identify five distinct stationary TME compositions that resemble the well-known, clinically verified TME subtypes, such as immune/fibrotic, immune/non-fibrotic, non-immune/fibrotic, and desert. Further, our investigation shows the immune/fibrotic composition can further be classified into two subcategories: fibro-dominated (fibro-rich) and immune-dominated (immune-rich). We observe that the quantitative balance between cancer-associated fibroblasts (CAF)-tumor interactions and the cytotoxicity of the killer T cell governs the post-ICI outcome in fibro and immune-dominated scenarios. While the immune desert TME, in most of the scenarios, remains insensitive to the ICI therapy, the immune-dominated HNSCC TME is characterized by a significant reduction of tumor cells and CAF compared to its pre-ICI population. The fibro-dominated TME, due to CAF-induced remodeling of the extracellular matrix, alters the accessibility of the killer T cells to the tumor cells. Our models suggest that a one-time IL-2-based intervention, beyond a critical T cell proliferation rate, can overcome the ICI insensitivity posed by an immune desert TME. We identify cytokines such as OPN and LIF as important targets to improve the ICI response of a fibro-dominated scenario. Finally, IL-8 and lactate show distinct signatures across different HNSCC TME subtypes identified by the model, indicating the potential to be considered essential biomarkers for therapy resistance (Table 1 provides a compact representation of the biomarkers and targets identified by the proposed model).

**Table 1. Potential biomarkers and targets identified by the model.** As predicted by the proposed model the IL-8 and Lactate can serve as prominent distinguishing factor between the resistant and non-resistant subtypes. However, OPN, LIF, and IL-2 can be appropriate target nodes for ICI resistance (non-response) emergent in the fibro-dominated and fibrotic only scenarios respectively.

| Identified TME subtypes | Biomarkers | | | | ICI outcome | Possible targets |
|---|---|---|---|---|---|---|
| | IL-8 | OPN | LIF | Lactate | | |
| Desert | Low | Low | Low | Moderate/ High | No response | One time IL-2 with Lactate knockout (KO) |
| Fibrotic only Exhaustion-driven | High | High | High | High | No response | One time IL-2+Lactate KO |
| Non-fibrotic | Low | Low | Low | Low | Response | – |
| Immune/fibrotic Immune dominated | Moderate | Moderate/ High | Moderate/ High | Moderate/ High | Response | – |
| Immune/fibrotic Fibro dominated | High | High | High | High | No response | OPN+LIF knockout |

This paper is organized in the following way: the 'Results' section begins with the primary goals of this modeling exercise and subsequently, the subsections lay out the important computational findings towards vis-à-vis the goals of this work. Fig 1 illustrates the overall workflow of the present study. The 'Discussion' section puts the model predictions in the context of the relevant clinical observations and experimental demonstrations in the literature. The 'Materials and Methods' section provides a detailed description of the model construction and necessary calculations for this work.

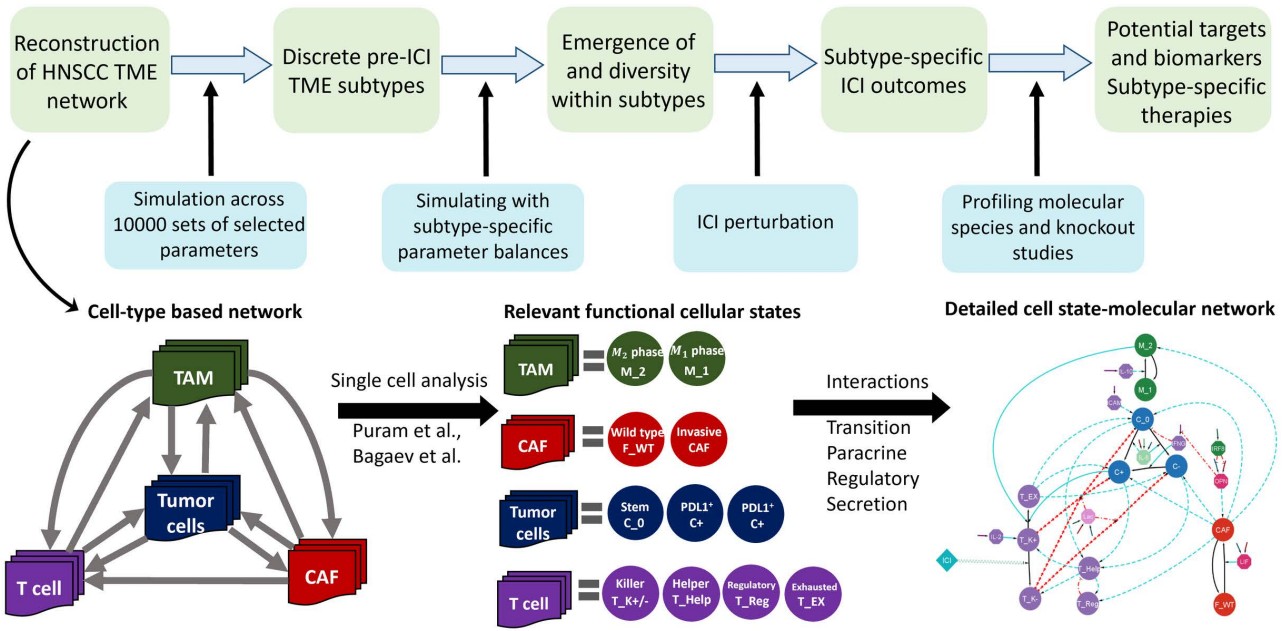

**Fig 1. Workflow: We start with construction of the essential cellular molecular network of the HNSCC TME from the existing literature.** For this purpose, we begin with the relevant cell types in a HNSCC TME obtained from the literature. Further, to incorporate a wide range of possible interactions from a given cell type, we considered different functional cell states within each cell type and the relevant molecular species mediating the cell-cell interactions. Using the well-established modeling rules for different types of interactions, we simulate the TME system with different combinations of selected parameters. The steady state populations can be grouped into five distinct compositional regions (in CAF-Killer T-tumor cell population space) resembling the clinically observed TME subtypes. We chose the relevant parameter balances within each compositional regions to unpack the different mechanisms that explains the emergence and diversity of a given TME compositional group. Further, we simulate the model with anti-PD1 for growth-proliferation parameters pertaining to each of the five compositional groups and important parameter balances constructed from the remaining parameter sets. Finally, we perform cytokine profiling and computational knockout studies to identify potential biomarkers and targets.

## Results

Based on the existing knowledge about the tumor biology of HNSCC and the modeling regimes, we frame the primary objectives of this work as the following:

1. Develop a whole tumor microenvironment-based mathematical model that provides a mechanistic mapping from the existing knowledge of cellular interactions to TME subtype diversity.

2. A model-based explanation of how the TME subtype diversity translates to prognostic heterogeneity in the context of ICI therapy.

3. Identifying the ICI-resistant TME subtypes and providing subtype-specific potential biomarkers and targets for circumventing ICI resistance.

For the purpose of model development, we reconstruct the overall TME network structure for HNSCC from an extensive survey of the existing literature [30–94]. Since the present work revolves around ICI treatment, we classify the tumor cells in three distinct tumor cell states. The stem-like tumor cells, albeit susceptible to the killer T cells, can transition to programmed death ligand-1 positive (PDL1+) tumor cells that can escape the immune onslaught [38]. The transition of stem cells into aggressive tumor cells have been shown to be facilitated by the tumor cell secreted IL-8 [38,39,52]. The stem-derived PDL1- tumor cells can further develop PDL1+program in the presence of IFNγ [36,57,77]. The overall proliferation of the tumor cells are also modulated via paracrine interactions by the cancer associated fibroblasts (CAF) [30,51,67]—a self-proliferating (in the presence of Osteopontin (OPN)) fibroblasts phenotype that is derived from the wild type fibroblasts in the presence of tumor cell and fibroblasts-derived leukemia inhibitory factor (LIF) [41,53,66,71,72,80–82]. On the other hand, the single cell RNA sequence (scRNA-seq) analysis of the HNSCC patients (primary tumor) by Puram *et al. (2017)* revealed the presence of CD8+cytotoxic killer T cells, CD4+helper T cells, CD4+CD25+regulator T cells, and CxCL13+exhausted T cells [34]. The PD1+killer T cells, in the presence of Interferon gamma (IFNγ), sense the tumor cells via the helper T cells and kill the PDL1- tumor cells [61,63]. On the contrary, the regulatory T cells inhibit the proliferation of helper T cells for regulating the overall immune response [63]. The PDL1+tumor cells can not only aid in immune escape but leads to the exhaustion of immune cells in the presence of M2 macrophages [46,59,60]. Interestingly, as demonstrated in the melanoma patients, the proliferation of the M2 macrophage is largely controlled by the CAFs via paracrine interactions [94]. We refer the readers to the section 'Reconstruction of the TME components and interactions' for a detailed discussion on model development.

With the modeling rules and assumptions declared in the methodology section, the proposed model develops into a dynamic system with twenty-four states and ninety-one parameters. The state variables denote the population (concentration) of the different cell states (molecular species). The assumption of spatial homogeneity translates to a system of ordinary differential equations (ODE). The mathematical expression of each flux and the systematic development of the model are presented in S1 Text and S2 Text, respectively.

### Classification of TME compositions

Given the proposed reaction atlas for the HNSCC TME (Fig 2(a)), and the associated dynamical system (refer to Materials and Methods section for details on model construction), we focus on identifying the possible TME subtypes that emerge from the model. For this purpose, we simulated the HNSCC TME system in Fig 2, with varying the parameters that have a direct influence on the proliferation and the death and conversion fluxes of the CAF, Killer T, and tumor cell population. The well-known phenomena of killer T cell exclusion by collagen disposition of CAF has been modeled using compartmentalization of the tumor cells into immune-accessible and CAF-protected modules (refer to Materials and methods section for detailed description). The remaining sixty-three parameters have been fixed to their nominal values (refer to S3 Text and S4_Text). The nominal values have been adopted from various sources The nominal values of all the parameters have been inspired from several sources including [15,23,24,113].

With ten thousand different realizations of the selected parameter sets (refer to S3_Text), we observe five distinct groups in the CAF-tumor-killer T cell landscape. Interestingly, the distinct groups identified in Fig 2(d-h) are consistent with recent work of Bagaev *et al.*(2021) regarding the characterization of the phenotypic heterogeneity of melanoma tumor microenvironment [4]. The non-desert TME compositions can have two different varieties, namely, fibro-dominated, and immune dominated. A fibro-dominated TME (group 5 of Figs 2(b) and 3(h)) is identified with an elevated level of CAF population and tumor cell population compared to the killer T cells.

Meanwhile, the immune-dominated TME lacks the population of PDL1 tumor cells and contains a strong presence of cytotoxic T cells (group 2 of Fig 2(b) and 2(g)). Interestingly, the immune and fibro-dominated TME contains similar levels of immune and CAF cells. Therefore, it is not the abundance of CAF and immune cells but the balance between the tumor-promoting role of CAF and the cytotoxicity of killer T cells that determines the tumor cell population, which serves as

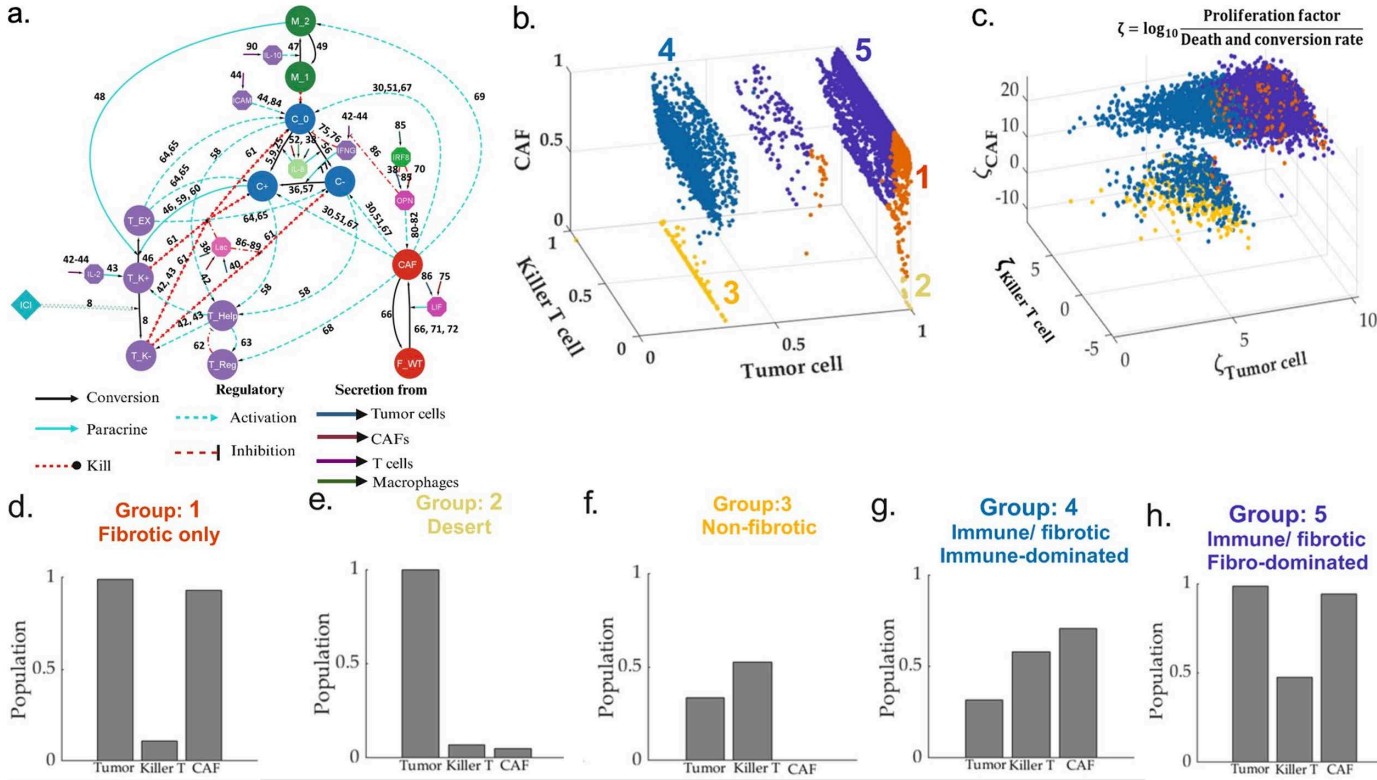

**Fig 2. Explaining the phenotypic heterogeneity of the HNSCC TME. (a)** The nodes are either the cell states or the molecular species, whereas the edges represent diverse forms of interactions. The acronyms C_0, C_PDL1+, and C_PDL1- refer to stem, PDL1+ (programed death ligand1), and PDL1- tumor cells, respectively. T_K+, T_K-, T_Help, T_Reg, and T_Ex stands for PD1+ (programmed death 1), PD1- killer T cells, Helper T cells, Regulatory T cells, and Exhausted T cells, respectively. M_1 and M_2 refer to macrophages of M1 and M2 phase, respectively. Further, F_WT and CAF correspond to wild-type and invasive cancer-associated fibroblasts, respectively. The acronyms IL-2, IL-8, IL-10 LIF, IFNG, IRF8, OPN, ICAM1, and Lac denote Interleukin-2, Interleukin-8, Interleukin-10, Leukemia Inhibitory Factor, Interferon Gamma, Interferon Regulatory Factor-8, Osteopontin, Intercellular Adhesion Molecule-1, and Lactate, respectively. Based on the accessibility of the tumor cells to the killer T cells, each of the three tumor cell states C_0, C+, and C- are further divided in to two categories Killer T-exposed and killer T non-exposed. S5 Text details all the abbreviations used in the network model. The killer T non-exposed tumor cell states are protected by the resident CAF population. (Refer to S1 Fig for detailed representation of the model) **(b)** We simulated the dynamics for 10,000 combinations of the parameters (that have a direct influence on the proliferation, conversion, and death of the CAF, Killer T, and tumor cells) based on the reconstructed network system to obtain the possible steady-state groups. We found five distinct groups: desert, immune desert, fibro desert, immune-dominated, and fibro-dominated. **(c)** The associated mapping of these five groups in the hyper-parameter space. **(d-h)** Bar charts for the median population of tumor cells, killer T cells, and CAF population across different groups.

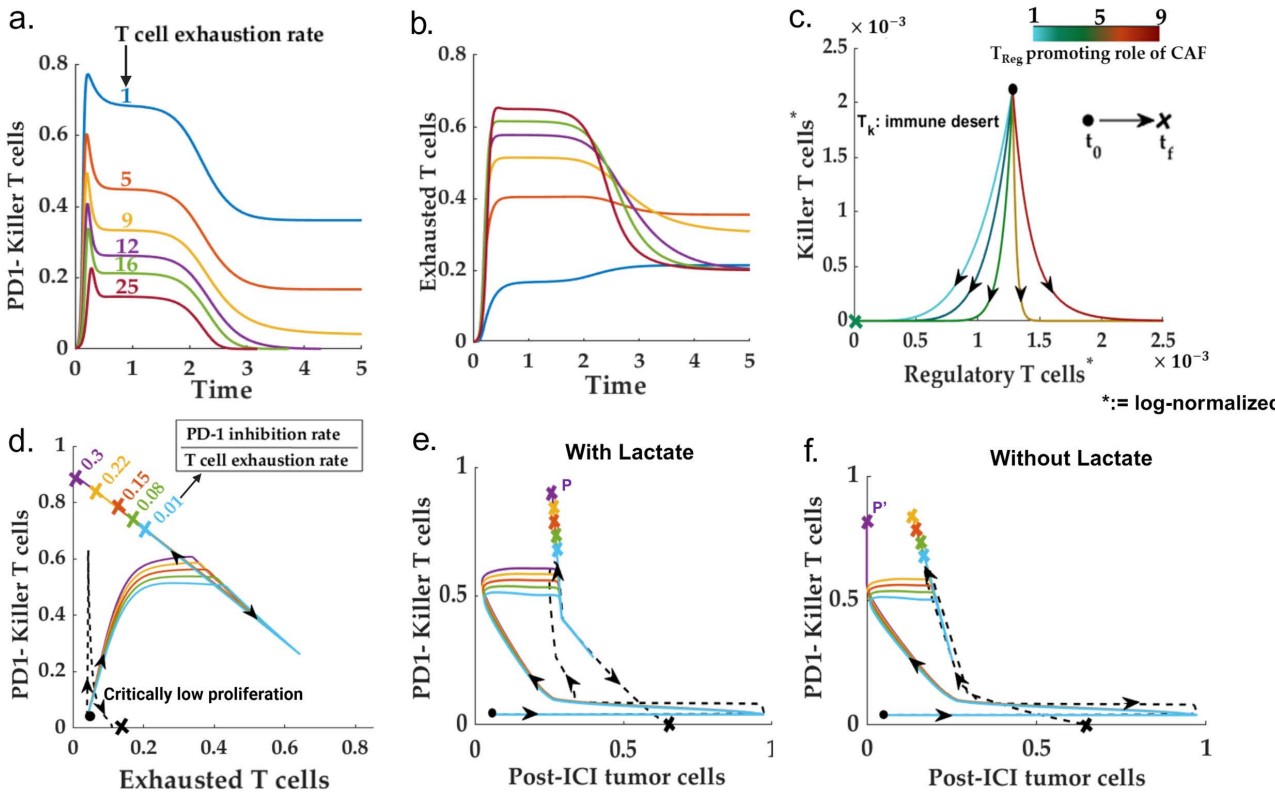

**Fig 3.** *The immune-desert scenario:* **(a-b)** Time profiles for killer and exhausted T cells for different T cell exhaustion rates. A high T-cell exhaustion program that can drive an immune system with a high proliferation rate to the immune-desert phenotype. **(c)** Phase space for killer T cells versus the regulatory T cells for different regulatory T cell promoting roles of CAF. Demonstrates the possibility of an immune-cold scenario with abundant regulatory T cells due to high CAF-Treg interaction. **(d)-(e)** Post-ICI trajectories of killer T cells, tumor cells, and exhausted T cells for different balances between the anti-PD1 binding rate and T cell exhaustion rate. Unlike other low-proliferation induced immune-desert phenotypes (represented in dotted lines), the ICI intervention can improve the exhaustion-driven immune desert scenario. **(f)** Lactate reduction can further reduce the total post-ICI tumor cell count.

the crucial distinguishing factor between the two TME phenotypes. Further, the ratio between the PDL1- to PDL1+ tumor cells can be a potential biomarker for distinguishing two distinct TME phenotypes.

On the other hand, the desert phenotypes are identified with zero or very few immune or (and) CAF cell population. Owing to the absence of the CAF population, the immune/non-fibrotic TME composition (group 3 of Fig 2(b) and 2(f)) is characterized by a lower tumor cell count compared to the immune desert scenario (group 1 of Fig 2(b) and 2(d)). We observe that although the immune-dominated, and immune/non-fibrotic, arrangements can be characterized with a lower tumor cell population compared to the fibro-dominated and immune-desert phenotypes, the HNSCC TME is never completely depleted of the tumor cells in the pre-ICI setting primarily due to the resource availability and immune escape and exhaustion mechanisms specific to each of the identified stationary TME compositions. This can be argued as the presence of IFNγ that, according to our model, plays a dual role of (a) enabling the cytotoxic T cell to recognize the PDL1- tumor cells and (b) induce PDL1 expression on the tumor cells that can aid in the immune escape. Although the prevalence of immune-desert TME subtypes spans across the human papillomavirus (HPV) status in the context of HNSCC, it is important to note the overall composition of the immune desert/ fibrotic only subtype (Fig 2(d)) strongly resembles the majority of the HPV-negative HNSCC cases where the underlying TME is depleted of the killer T cells.

## Diversity within the immune desert

We observe that the immune desert and desert (Group 1 and 2 of Fig 2(b)) contributes as the primary source of high PDL1- tumor cells. Interestingly, even for a significant fraction of the fibro-dominated scenario, the final killer T cell population settles to extremely low levels despite a high proliferation-to-death balance. Further, several scenarios pertaining to the immune desert setting reveal a moderate to high exhausted and regulatory T cell population. Therefore, we simulated the TME system for varying proliferation and exhaustion rates of killer T cells to understand the diverse mechanisms of immune depletion. Our analysis reveals that apart from a critically low proliferation rate, a strong T cell exhaustion program may lead to an immune desert phenotype (Fig 3(a-b)). Interestingly, a specific scenario may mimic the immune desert phenotype wherein the immune system is composed of many regulatory T cells with almost no cytotoxic or helper T cells. We observe that a higher paracrine interaction rate between CAFs and the regulatory T cells can trigger the immune-cold scenario despite a low proliferation-to-death ratio for the regulatory T cells (Fig 3(c)). This is in line with the recent findings of Bao *et al.* obtained through the single cell-RNA seq analysis of the colorectal cancer patients wherein a subset of anti-PD1 (nivolumab) non-responders presented with high regulatory T cells, and very low CD8 + killer T cells and CD4 + effector (helper) T cells-- indicative of an immune-cold situation [93].

Overall, the proposed model identifies at least five different TME subtypes namely, immune desert or fibrotic, desert, non-fibrotic, immune dominated, and fibro-dominated. Interestingly, each of these five TME subtypes have been observed in HNSCC, melanoma, or colorectal cancer patients [4,96,99]. Table 2 provides a consolidate representation of these results vis-à-vis the supporting evidences in the existing literature. Subsequently, we leverage the proposed model to examine the ICI response for each of these five subtypes.

## Effect of ICI: specific to TME subtypes?

As discussed, none of the identified stationary TME compositions leads to a complete depletion of tumor cells due to the assumption of high tumor cell proliferation rate compared the death rate. The PDL1 + tumor cells, unlike the stem and PDL1- tumor cells escape the innate and adaptive immune response in the pre-ICI setting via PD1-PDL1 binding and consequent immune exhaustion [46,59,60]. The resultant PDL1 + tumor cell population remains non-zero as long as (a) the self-proliferation rate is even slightly higher than the death rate and (b) the initial population is non-zero—this necessitates an ICI-based intervention (anti-PD1) that eliminates the PD1 program in the killer T cells. The killer T cells without the PD1 program (PD1- killer T cells) can recognize PDL1 + tumor cells as antigen presenting cells. We studied the role of anti-PD1 dosage for all the different parametric conditions (Fig 2(c)) mapping to the desert (cold) and dominated TME phenotypes.

## Are all immune deserts non-responsive to ICI?

As discussed previously, multiple mechanisms exist (low-proliferation, exhaustion-driven, and $T_{reg}$ abundant) behind the emergence of an immune desert HNSCC TME (Fig: 3(a-c)). To investigate whether the ICI response to the immune desert scenario is dependent on the specific desert mechanism, we simulated the proposed system with a non-zero anti-PD1 level for different variants of parametric conditions reflective of all the possible groups within the immune-desert phenotype in the pre-ICI scenario. To resemble a realistic scenario, the killer T cell population and IL-2 concentration are kept non-zero in the initial condition for the ICI simulation (Fig 3(d)). As observed in Fig 3(d), ICI intervention does not impact the progression trajectory of low-proliferation driven immune desert TMEs (dotted lines in Fig 3(d)). We observed that unlike a low-proliferation caused immune desert, an exhaustion-driven desert can be responsive to ICI therapy. On the contrary, an exhaustion-driven TME can have increased killer T cell recruitment following the anti-PD1 treatment. However, as suggested in Fig 3(d)-3(e), this response is governed by the balance between PD1 inhibition rate and T cell exhaustion rate. Further, in Fig 3(f), our model predicts that a Lactate knockout may further improve the ICI response of an exhaustion-driven immune desert and can lead to complete removal of the tumor cells beyond a threshold

**Table 2. Comparing the model conclusions with the appropriate existing literature.**

| Model predictions | Validating reference | Description |
|---|---|---|
| 1. Classification of five distinct TME subtypes | 1. Bagaev *et al.* (Cancel cell, 2021) [4]<br>2. Quek *et al.* (Cell reports, 2024) [99]<br>3. Kech *et al.* (Clinical cancer research, 2015) [96] | 1. TME classification via the transcriptomic analysis of 10000 different cancer patients. The single-cell data has been obtained through the TCGA dataset<br>2. The single-cell omics analysis of 40 metastatic melanoma patients revealed the immune-desert and fibro-rich TME subtypes have a strong correlation with the outcome of immunotherapy.<br>3. With a set of tools pertaining to genomics and copy number profiling of 134 metastatic HNSCC patients with 44% HPV signature, Kech *et al.* identified immune-rich (prominent CD8＋T cell infiltration in the tumor), fibro-rich (hypoxic) and immune desert (low CD8＋expression) TME subtypes. |
| 2. Diversity within immune desert: Exhaustion-driven and immune-cold (Prevalence of Tregs) | 1. Bao *et al.* (Cell reports medicine, 2024) [98]<br>2. Wang *et al.* (British journal of cancer, 2023) (Review on Treg-driven and TExh-driven desertification of Killer T cells in TME) [74]<br>3. Farlow *et al.* (Oral oncology, 2021) (Review on immune-desert for HNSCC) [101] | 1. Single-cell analysis of the TME for colorectal cancer from the TCGA dataset reveals the existence of TME subtypes with CD4＋regulatory T cells, CD8＋exhausted T cells, and extremely low CD8＋Killer T cells. |
| 3. Immune desert (fibrotic only) and fibro-dominated TME subtypes are resistant to anti-PD1-based therapy | 1. Quek *et al.* (Cell reports, 2024) [99]<br>2. Estephan *et al.* (Frontiers in oncology, 2024) [102]<br>3. *Xiao et al.* (Nature communications, 2023) [103] | 1. Existence of a TME variant for melanoma TME with few Killer T cells excluded from the tumor– Possible fibro-dominated scenario.<br>2. A subset of a total of 121 patients with Oral Squamous Cell Carcinoma presented with a TME non-respondent to immunotherapy that contains an altered tumor cell position with respect to the CD8＋T cells due to modified extracellular protein secreted from specific subtypes of CAF.<br>3. The existence of fibroblast activating protein(FAP+) CAFs in desmoplastic pancreatic tumors leads to treatment-resistant TMEs by physically excluding the CD8＋killer T cells from the solid tumors. |
| 4. Increase in IL-8 in Fibro-rich non-responders | 1. Hill *et al.* (2023, Molecular Carcinogenesis) [93] | 1. Post-ICI (anti-PD1) plasma IL-8 follows a slight increase compared to pre-ICI for HPV-negative (low immune-accessibility according to our model) non-responders of HNSCC patients– as identified through an extensive tissue analysis of institute cohort HNSCC patients undergone neoadjuvant ICI therapy with nivolumab. |
| 5. One-time IL-2 spike: Retrieve ICI-sensitivity of an immune-desert subtype. | 1. Bright *et al.* (Journal of immunotherapy, 2017) [104]<br>2. *Mitro* et al. (Science translational medicine, 2022) (Reviews the existing IL-2 therapies in Cancer immunotherapy.) [105]<br>3. Muhammad *et al.* (BMC molecular cancer, 2023) [106] (Perspective on using a one-time IL-2 shot to bring back the ICI sensitivity) | 1. Summary of 3312 metastatic melanoma patients treated with IL-2. Suggestive of the fact that IL-2 works best in a moderate amount for an immune dessert (non-fibrotic/ immune cold in the terminology of our work) tumor microenvironment. |
| 6. OPN＋LIF knockout modifies the fibro-rich TME towards ICI sensitivity | 1. Albrengues *et al.* (Nature communications, 2015) [69]<br>2. Weber *et al.* (Oncogene, 2015) [107] | 1. *In vitro* and *in vivo* models of breast head and neck carcinomas, the wild-type fibroblasts convert to a pro-invasive CAF that is responsible for ECM remodeling and immune suppression. Further, the knockout of LIF brings back the wild-type prevalence^*.<br>2. In vitro studies of the human mesenchymal stem cells and OPN+ human breast cancer cell lines confirm that OPN increases the conversion of CAF from stem-like mesenchymal cells in breast cancers^*. |

^*: These references, unlike others, do not validate our prediction. It demonstrates the role of the selected target species in tumor progression, thereby justifying our selection. In these aspects, our model makes testable predictions on the effect of the knockout of LIF and OPN in a Fibro-rich scenario.

balance of PD-1 inhibition rate and T cell exhaustion rate. However, as evident from the dotted lines in Fig 3(f), the low-proliferation-driven immune desert remains insensitive to ICI even after Lactate knockout. Moreover, our findings suggest that the responsiveness of immune deserts to ICI therapy is not uniform and depends on the specific desert mechanism. This understanding can potentially guide the development of more personalized and effective ICI therapies.

### Immune/non-fibrotic presents a favorable scenario for ICI

The immune/non-fibrotic scenario is identified by the parametric conditions that leads to very low CAF population in the pre-ICI setting (Group 3 of Fig 2(b)). The pre-ICI, low CAF population rules out the possibility of the formation of immune-inaccessible pockets inside the TME. Further, due to the absence of the most dominating interaction from CAF to TAM, a immune/non-fibrotic scenario may also be identified with a low M2 macrophage population [94]. Therefore, the primary TME constituents impacting tumor cell growth are the immune cell states. For this scenario, we simulated the proposed model for varying rates of oncogenic roles of exhausted tumor cells and killer T cell cytotoxicity. A high ratio of the interaction rate between the exhausted T cells and tumor cells to the cytotoxicity of the killer T cells enhances the total tumor cell population in the pre-ICI setting. Further, this increase in the total tumor cell population is fueled by the increase in the population of PDL1+ tumor cell population (Fig 4(a-b)).

The ICI intervention predictably yields an improved prognosis with a significant shrinkage in the tumor cell population. With a prominent level of cytotoxicity, the anti-PD1 intervention can lead to a zero exhausted T cell population (Fig 4(c)). Application of anti-PD1 favors the balance towards the tumor suppression, thereby steering the resultant total tumor cell population towards zero. Although anti-PD1 yields a favorable outcome for the immune/non-fibrotic scenario, our simulation indicates that the post-ICI scenario does not necessarily lead to a zero-tumor cell population for high resource availability (Fig 4(d)) --- This can lead to the recurrence of the disease in a post-ICI scenario. Note that due to the absence of CAFs, the growth of tumor cells occurs at a much longer timescale when compared to a fibro-dominated situation.

### ICI reduces the invasive CAF population in immune-dominated HNSCC TME

As discussed in the previous sections, the immune-dominated TME subtype can be identified with an abundant killer T cell population and a low PDL1- to PDL1+ tumor cell population (group 4 in Fig 2(b)). Therefore, the primary potential factors governing the growth, immune escape (in pre-ICI condition), and the efficacy of the ICI treatment are the killer T cells, exhausted T cells, and resident CAF population. To determine the pro-tumor components' dependence in the HNSCC TME on the resident killer T cell population, we simulated our model with an increasing killer T cell proliferation rate.

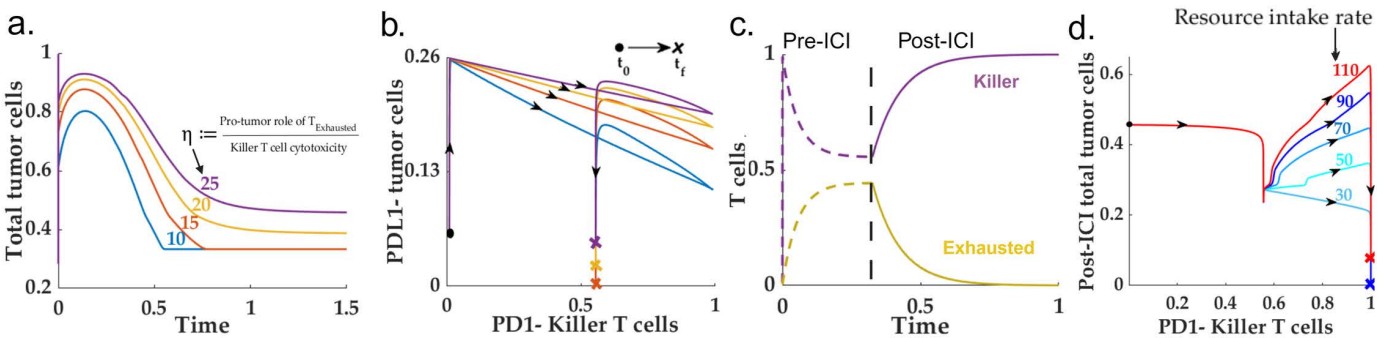

**Fig 4. *Immune/non-fibrotic, favorable case:* (a-b)** Shows despite the absence of CAF, the pre-ICI PDL1- tumor cells critically depend on the balance between the oncogenic (via Texh) and cytotoxic activities of T cells. **(c)** The intervention of ICI drives toward an aggressive cytotoxic T-cell-driven immune response. **(d)** Indicates the pivotal role of residual fixed resource supply rate (nutrients, blood flow, oxygen supply) in governing the possibility of recurrence despite a favorable prognosis.

Interestingly, we found out that beyond a critical proliferation rate, the final population of killer T cells does not vary significantly with respect to the proliferation rate (Fig 5(a)). This is due to the insensitivity of the CAF population to the proliferation rate of the killer T cells (S2 Fig). Further, the CAF population remains unchanged across the killer T cell proliferation rate—This may explain the CAF abundance in an immune-dominated TME. On the other hand, for high levels of cytotoxicity, the balance between the tumor cells vis-a-vis the PDL1 signature remains heavily skewed towards the PDL1 + tumor cells (Fig5(b)). Additionally, in resource competition environment, high levels of resource availability can be identified with high helper T cell counts owing to the abundance of PDL1- tumor cells. Our simulation suggests existence of a threshold maximum resource holding beyond which the helper T cell population becomes independent of the balance between the opposing roles of CAF ($\Gamma$) vis-a-vis the helper and killer T cell population (S3 Fig).

To determine how the exhaustion of the T cell interrupts the immune response in a pre-ICI setting, we simulated the immune-dominated TME system with different cytotoxic levels and repeated the exercise for different exhaustion rates. Interestingly, the PDL1- -tumor cell population critically depends on the exhaustion rate of the killer T cells, but in the

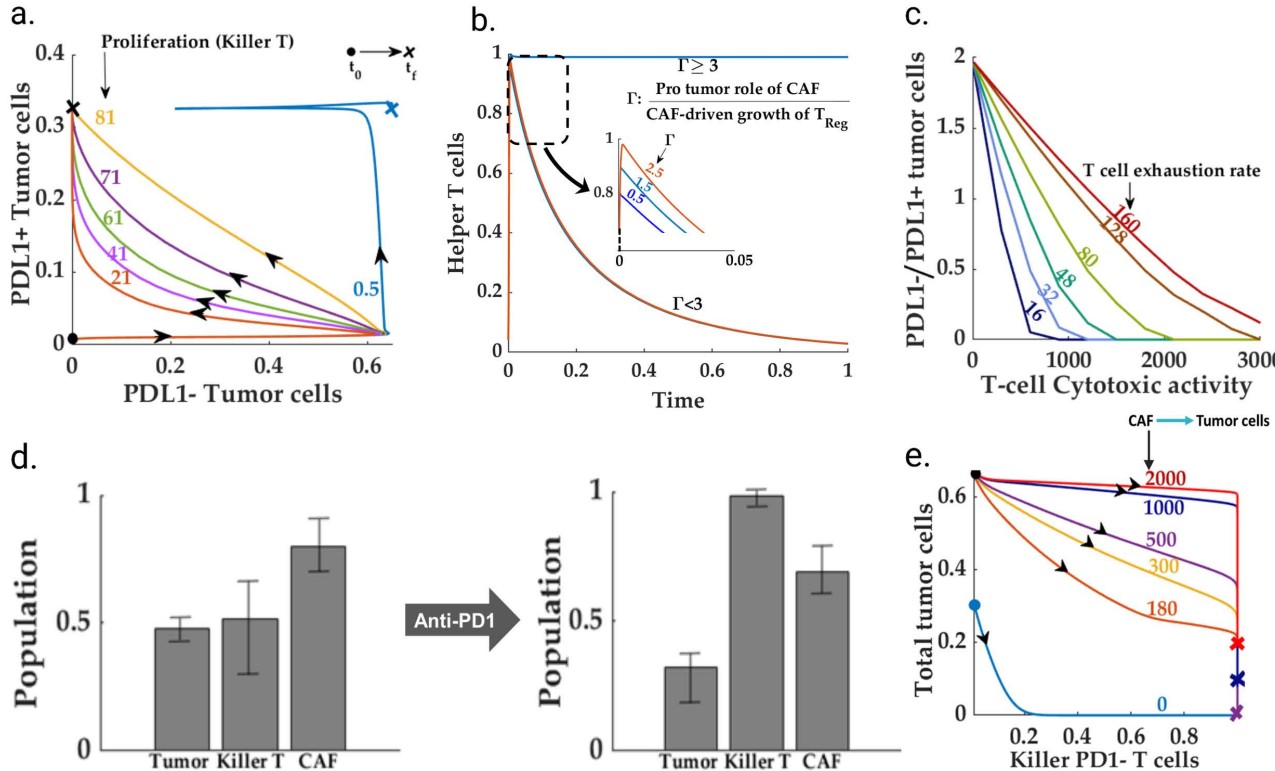

**Fig 5.** *Immune-dominated-- High immune accessibility, moderate CAF:* **For a given CAF-tumor interaction rate, lower T-cell cytotoxicity leads to higher PDL1⁻ to PDL1⁺ tumor cell population.** **(a)** For high levels of cytotoxicity, the tumor cell population remains insensitive to the proliferation rate ($K_{TK}^{prol}$) of the killer T cells in the immune dominated scenario. **(b)** CAF plays a dual role in governing the helper T cell population. CAF reduces the helper T cells via the regulatory T cells. However, a high tumor-promoting role of CAF increases the resident PDL1- -tumor cells that, in turn, increases the helper T cells via antigen sensing mechanism. **(c)** A high exhaustion rate can potentially increase the share of the PDL1- tumor cell population owing to the abundance of the exhausted tumor cells and their tumor-promoting effects. **(d)** The post-ICI scenario significantly reduces the tumor cells and CAF. The application of anti-PD1 reduces the resident CAF population and drastically reduces the tumor cell population. The population refer to the median population in the immune-dominated group (S6 Fig). Furthe the line bars indicate the 25% and 75% quartiles of the population within the immune-dominated scenario. **(e)** The post-ICI tumor cell population is proportional to the tumor-promoting role (via paracrine interaction) of the remaining CAF population.

presence of highly cytotoxic killer T cells, the pre-ICI tumor cell population does not possess any significant dependence on the proliferation rate of the killer T cells (Fig 5(c)).

To examine the efficacy of an ICI-based treatment for immune-dominated subtypes, we selected 10,000 different parameter sets corresponding to the immune-dominated subtype. We simulated both the pre-ICI and post-ICI scenarios. We observed that for high immune accessibility, the immune-dominated TME subtypes significantly reduce the total tumor cell population along with the CAF population (Fig 5(d)). Although the post-ICI outcomes for immune-dominated phenotypes can be identified with a significant reduction in the CAF or, equivalently, an increase in the wild-type fibroblasts, the chance of recurrence persists due to non-zero CAF-tumor cell interaction flux owing to non-zero OPN and CAF levels (Figs 5(eS4). Therefore, targeting the CAF population can also be a potential therapeutic strategy in an immune-dominated TME.

### Fibro-dominated: Effective immune desert?

Unlike the immune-dominated subtypes, the fibro-dominated TME is identified by exceedingly high CAF-tumor cell paracrine rates, low immune accessibility, and an abundant CAF population. As illustrated in Fig 6(A), an HNSCC TME with low immune accessibility index can be conceived as dividing the TME locally into two distinct regions vis-a-vis the tumor cells—1) immune-accessible tumor cells and 2) CAF-protected tumor cells (Fig 6(a)).

To find out the dependence of the overall ICI outcome on immune accessibility, we simulated the fibro-dominated HNSCC TME model with different immune accessibility indices. We observed the ICI dose response deteriorates linearly with the reduction of immune accessibility. Further, the balance between the anti- and pro-tumor roles of the T cell states modulates the slope of the dose efficacy-immune accessibility curve (Fig 6(b)). Further, unlike the immune-dominated scenario, the post-ICI CAF population remains almost identical to the pre-ICI counterpart in the low immune-accessible scenario (Fig 6(d)). The invariance of the CAF population to ICI in the fibro-dominated setting is one of the most critical prognostic differences between the immune-dominated and fibro-dominated TME subtypes. A simulation study of the CAF-immune accessibility index for multiple CAF proliferation rate reveals that the immune accessibility index increases gradually (smoothly) with respect to the growth of CAF (S5(a)) Fig. However, there exists a threshold timepoint dependent on the CAF proliferation rate, beyond which the immune accessibility deteriorates drastically (S5(b) Fig). To understand the mechanism, we studied the phase space in the LIF-tumor cell plane. Unlike the immune-dominated conditions, we observed that lower immune accessibility does not significantly change the LIF levels--- a significant player in the

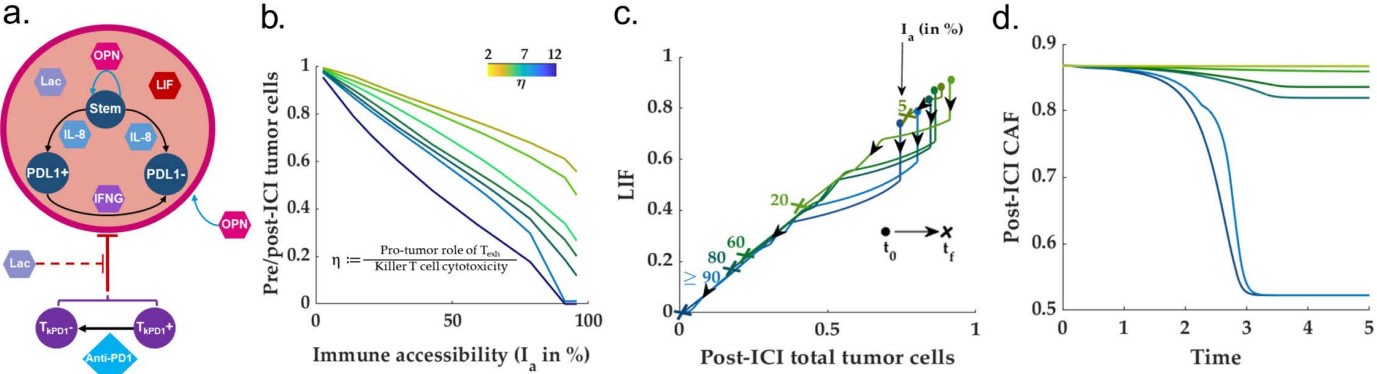

**Fig 6. Fibro dominated, a modified immune desert: (a)** depicts the conceptual framework of the proposed hypothesis on the CAF-pockets protected from the immune response. **(b)** demonstrate the fact that the immune accessibility index governs the post-ICI proportion of immune-accessible and immune-inaccessible tumor cells. **(c)** Phase-space for LIF and total tumor cell population. The change in LIF levels due to the ICI therapy reduces with respect to decreasing accessibility. **(d)** Contrary to the immune-dominated TME, the post-ICI CAF population remains almost constant at low immune accessibility due to a lack of change in the post-ICI LIF levels compared to its pre-ICI counterpart.

conversion from the wild-type fibroblasts to CAFs, thereby retaining the pre-ICI conversion flux (Fig 6(c)). The abundance of CAFs in ICI non-responders can also be verified from the experimental observations from melanoma, oral squamous cell carcinoma, and other solid tumor patients wherein the application of anti-PD1 did not result in any substantial reduction in tumor cells due to altered tumor cell position with respect to CD8+T cells owing to the CAF-induce modification of the extracellular matrix in Oral Squamous cell carcinoma [102]. Further, the existence of fibroblast activating protein+ (FAP+) CAFs correlates with ICI-resistance in pancreatic tumors [103].

In summary, based on the model prediction, the low-proliferation immune desert and fibro-dominated TMEs do not improve post-ICI intervention in a significant manner. On the other hand, the non-fibrotic, and immune dominated subtypes remain sensitive to ICI therapy thereby indicating the potential of improved prognosis. Therefore, in what follows, we focus on the target species, and possible biomarkers for the ICI-resistant TME phenotypes. This also motivates the interesting question of whether both the TME subtypes (immune-desert and fibro-dominated) require similar combinational therapies to improve the ICI response.

**Leveraging the molecular landscape of the TME- possible targets and biomarkers.** According to the proposed model, the post-ICI outcomes for different pre-ICI TME subtypes are significantly distinct. Although the fibro desert and immune-dominated TME subtypes result in the shrinkage of the total tumor cell population, the immune-desert, and fibro-dominated TME subtypes remain largely insensitive to ICI-based treatment. Therefore, to circumvent the ICI insensitivity emergent in some TME subtypes, we explore the prospect of modulating the well-known molecular species in the HNSCC TME. The no-show of ICI response for immune-desert TMEs can be attributed to the zero (or very few) pre-ICI killer T cell population. Therefore, any combination therapy in an immune desert scenario should aim to modulate the proliferation to death and conversion balance for the killer T cell ($\zeta_{Killer\,T\,cell}$). For this purpose, we chose IL-2 to enhance the killer T cell population in an immune desert scenario.

On the other hand, the insensitivity to ICI in the fibro-dominated parameter regions emerges due to either a high CAF-tumor cell paracrine rate or a low immune accessibility index. The common bottleneck in both scenarios is the abundance of the CAF population. Therefore, targeting the CAF via selected molecular species may improve immune accessibility and reduce the CAF-tumor paracrine rate. For this purpose, we selected OPN and LIF as the target molecular species for both molecules, which play a crucial role in the proliferation of- and conversion to CAF.

ICI+IL-2, reprograms the immune-desert TME As discussed in the preceding sections, a significant fraction (due to low Killer T cell proliferation) of the immune-desert TME subtypes do not respond to ICI treatment. The primary reason is the depletion of killer PD1−killer T cells under pre-ICI conditions. Further, due to the small killer T cell population, the final, pre-ICI, IL-2 concentration settles to an extremely low level. According to the HNSCC TME network, the population of the killer T cells can be enhanced through IL-2. Therefore, we simulated an immune-desert TME with one-time, initial IL-2 spikes of varying amplitude. Interestingly, we observed that for one time, IL-2 spikes beyond a threshold level, the cell state population trajectories move away from the immune-desert to an immune-non-desert trajectory (Fig 7(a-c)), which renders a favorable post-ICI outcome for a high immune accessibility index. The threshold one time IL-2 levels required to drive the TME away from immune desert is dependent on the balance between the IL2-driven autocrine and exhaustion, death rate of killer T cells (S6 Fig).

## Restoring sensitivity to ICI for immune-inaccessible, fibro-dominated TME

According to the proposed network, the cancer-associated fibroblasts grow through i) self-proliferation via OPN-mediated autocrine interaction and ii) LIF-induced conversion from wild-type fibroblasts. Therefore, we study the effect of OPN and LIF reduction on the ICI outcome for a fibro-dominated TME with very low immune-accessibility. To understand how the balance between the immune-accessible and immune-inaccessible tumor cells depends on the OPN level, we simulated the proposed HNSCC model for different OPN reduction rates. We repeated this exercise in the presence of LIF and with LIF being knocked out. We observe that the reduction of OPN modifies the balance between the tumor cells vis-à-vis immune accessibility towards the accessible tumor cells. Further, a LIF knockout and OPN reduction significantly

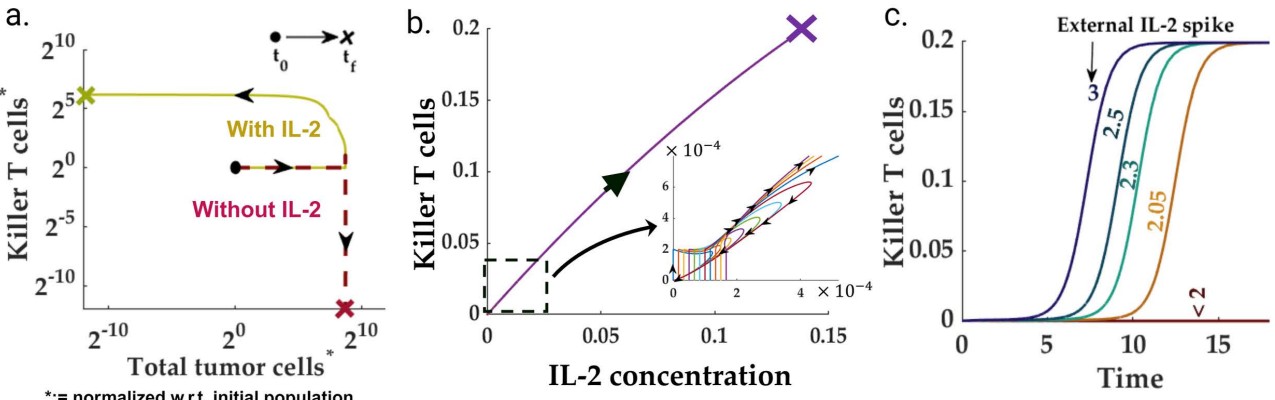

**Fig 7. *One-time IL-2 spikes— reprogramming immune-desert*: (a)** A spike in the initial IL-2 levels (beyond a threshold) can drive the system trajectories towards an immune-non-desert steady state. **(b)** Although some trajectories return to the immune-desert arrangement, the trajectory settles in a non-zero killer T cell population beyond a threshold IL-2 injection. **(c)** Time profiles for killer T cells for different external IL-2 levels.

improved over both the OPN-based treatment (Fig 8(a-b)) and LIF-knockout (S8 Fig). Interestingly, the tumor cell balance reduces drastically beyond a parameter-dependent threshold anti-OPN concentration. Moreover, the reduction in the total CAF count leads to an improvement in immune accessibility, thereby bettering the chances of recovery (Fig 8(c)).

In summary, the potential combination therapies for immune-desert and fibro-dominated are clearly different. Although the HNSCC TME subtypes produce similar diagnoses (barring the total T cell counts in the TME) vis-à-vis different tumor cell state populations, the IL-2-based treatment in the immune desert focuses on strengthening the immune system. In contrast, the OPN+LIF knockouts target the CAF population in the TME to improve the immune accessibility landscape. In the next section, we leverage our model to identify the possible biomarkers that can indicate the existing tumor micro-environment subtype and therefore can aid in determining the appropriate combination therapy.

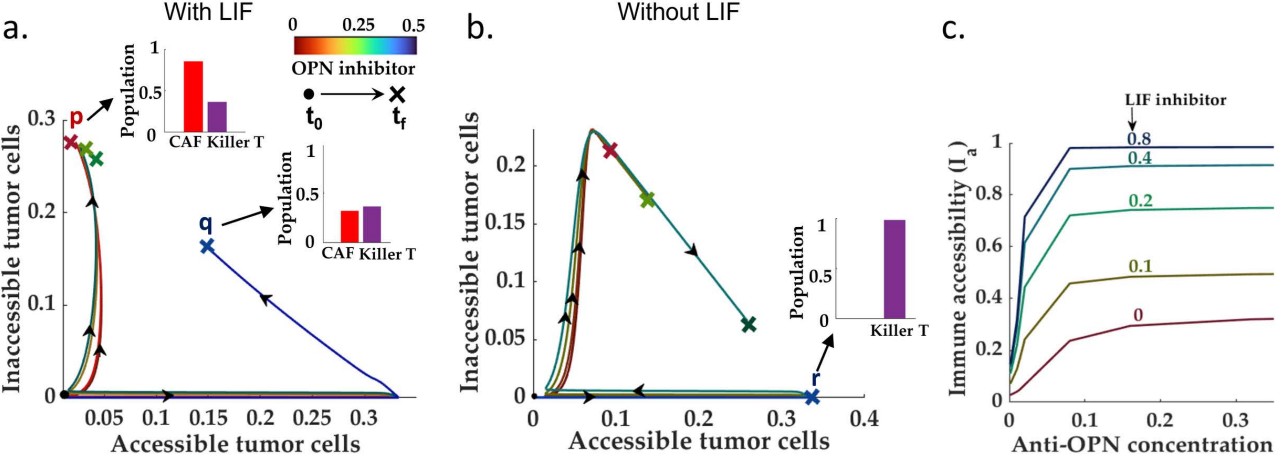

**Fig 8. *OPN and LIF knockout, from fibro-dominated to fibro-desert*: (a-b)** Demonstrate the effect of OPN reduction on the proportion of the inaccessible to accessible tumor cells in two scenarios: with and without LIF. As shown in both cases, below a certain threshold of OPN concentration, the balance between the accessible and inaccessible tumor cells improves significantly. An additional knockout of LIF extends the OPN-knockout-driven reduction in tumor cells towards complete removal. **(c)** Suggests that OPN reduction modulates the immune accessibility index of the TME. Further, a LIF+OPN reduction drives the TME towards full accessibility.

## IL-8: Primary distinguishing factor for different TME subtypes

As established in the preceding sections, TME subtypes play a governing role in determining the post-ICI outcomes. Unlike other cellular species, most molecular species do not exhibit specific signatures unique to TME subtypes. For instance, the IL-2 levels remain similar in both the immune-dominated and fibro-dominated scenario for the resident killer T cell population, which does not present significant variation between immune-dominated and fibro-dominated subtypes (Fig 2(b-h)). The fold change of the OPN levels (pre- vs. post-ICI) remains indistinguishable for fibro-dominated and immune-desert scenarios for the post-ICI CAF population. It does not undergo significant changes in the TME subtypes (Fig 6(c)). The scenario for LIF is similar (Fig 6(d)). Therefore, we set out to investigate the prospect of IL-8 as a potential biomarker for TME subtypes.

Interestingly, we found out that IL-8 exhibits distinct trajectories for different TME subtypes (Fig 9(a-b)). In the immune desert scenario, due to the absence of effective killer T cells, the post-ICI IL-8 does not change. Although the immune/non-fibrotic and immune-dominated TME subtypes are marked by a significant decrease in the IL-8 levels, the absolute levels of IL-8 concentration are significantly different for both scenarios. More importantly, unlike other inflammatory cytokines such as OPN and LIF, the IL-8 level shows a slow *increase* in the post-ICI scenario for an immune inaccessible fibro-dominated scenario. This can be verified by the recent experimental observations by Hill *et al.* (2023) in HPV-negative HNSCC patients. As reported in the recent study by Tosi *et al.* (2022), the HPV negative scenario is largely identified with low immune infiltration- potentially resembling the low immune accessibility condition according to our model. The IL-8 levels in the plasma slightly increase for non-responsive HPV-negative patients compared to its pre-ICI counterpart.

## Lactate reduction improves the overall ICI response

Lactate is a well-known metabolite reported in the literature on several solid tumors, including HNSCC. Interestingly, a non-responsive TME has been identified with elevated lactate levels, whereas a good prognosis, in some scenarios, has been identified with lower lactate levels [91].

On the other hand, to study the lactate signature for different immune accessibility scenarios, we simulated the system for various values of immune accessibility. We chose the model parameters so that high immune accessibility resembles an immune-dominated scenario. We obtained the phase space for post-ICI lactate and tumor cell populations.

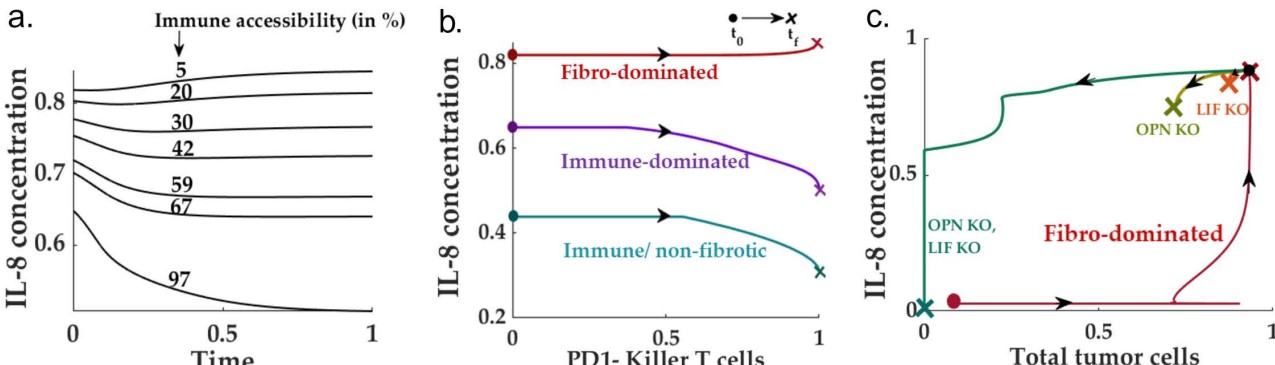

**Fig 9. *IL-8: Potential biomarker for identifying TME subtype*: (a-b)** Show that with high immune accessibility, the IL-8 level is low and is further reduced following a neo-adjuvant ICI therapy, whereas the scenario with low immune accessibility (characteristic of a fibro-dominated TME) can be identified with a slight increase in the post-ICI IL-8 level. Although the IL-8 trend is similar for immune-dominated and immune/non-fibrotic TME, the absolute levels differ significantly between the two subtypes. **(c)** The removal of both OPN and LIF reprograms the TME towards a immune/non-fibrotic subtype. Therefore, the IL-8 level post-OPN and LIF knockout followed by ICI also drops drastically compared to its pre-ICI and pre-removal counterparts.

Interestingly, the ICI intervention reduces the residual lactate levels at most immune accessibility, barring extremely inaccessible situations (Fig 10(c)). Interestingly, the post-ICI lactate level slightly increases from its pre-ICI counterpart. This suggests the potential use of lactate as a biomarker for low immune accessibility.

We leverage the proposed model to understand the effects of lactate on the population of tumor and non-tumor cells. For this purpose, given a specific set of parameters, we calculated the proportion of PDL1- to PDL1+ tumor cells for varying killer T cell cytotoxicity levels (Fig 10(a)). We repeated the simulation for various residual lactate levels. We observed that an increasing amount of lactate pushes the TME away from immune-dominated to an effective immune-desert (or CAF-dominated) scenario (Fig 10(b)). Therefore, targeting lactate can increase the killer T cell efficiency and move the TME to an immune-dominated region for moderate to high immune accessibility.

## Discussion

In this work, we developed a mathematical model for the HNSCC TME that explains a range of clinically (and experimentally) observed phenomena ranging from multiple TME subtypes to distinct responses to immunotherapy. The conclusions drawn from the extensive simulation studies of the model are placed in the context of the existing literature (95–115). Table 2 consolidates and contextualizes some of the key findings vis-à-vis the recent clinical and experimental observations.

Considering the trade-off between the ease of computation (theoretical/ computational analysis) and the granularity (explainability), we chose different transcriptomic cell states (and molecular species) as the agents of the proposed model. It enabled us to work with a smaller dimensional system (compared to gene regulatory network formalism) and without significant compromise on explainability. With an exhaustive search across the parameter space, we identified five distinct TME compositions, each representing clinically observed TME subtypes: fibro-dominated, immune-dominated, immune/non-fibrotic, immune-desert, and desert [4,96,97,99]. Our model suggests that different parameter balances determine the possible path of the cell state population trajectories toward a particular TME phenotype. Whether the exact factors that govern these balances may be epigenetic or owe their emergence to mutation remains an open area of study. We observe these pre-ICI TME compositions govern the post-ICI response. Our model predicts that while the immune/non-fibrotic and immune-dominated HNSCC subtypes constitute a favorable situation for ICI therapy, immune-desert and fibro-dominated

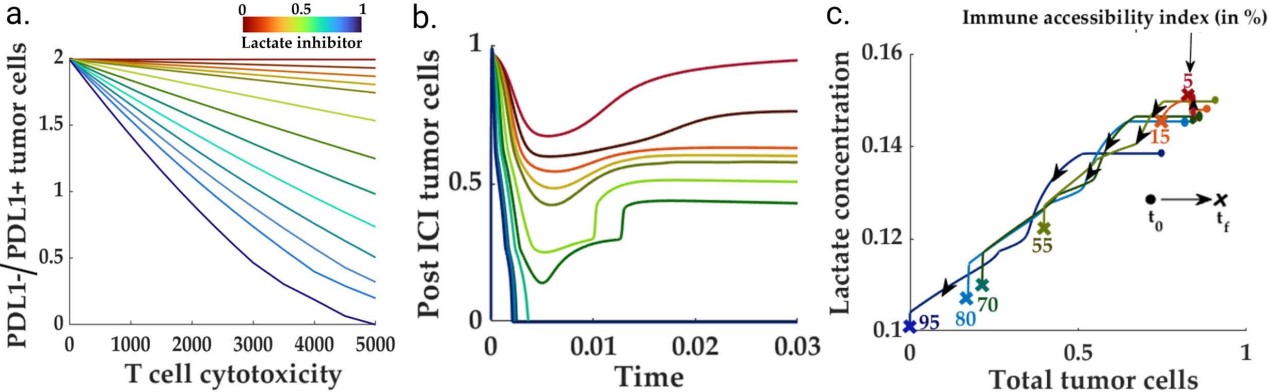

**Fig 10. *Lactate removal improves overall immune response:*** **(a)** Lower levels of residual lactate concentration inside the TME enhance the cytotoxicity of the killer T cells in the pre-ICI setting. **(b)** The negative impact of lactate on the immune response continues to the post-ICI scenario, wherein a high lactate level impedes the PD1-killer T cell activity, leading to a high tumor cell population. **(c)** For high-killer T cell cytotoxicity, lactate can also serve as a distinguishing factor between immune-accessible and inaccessible HNSCC TMEs. In highly immune-accessible TMEs, the application of anti-PD1 renders a significant decrease in lactate compared to its pre-ICI counterpart. In contrast, the low immune-accessible TME correlates with higher post-ICI lactate levels.

TMEs remain insensitive to therapy resistance for a wide range of model parameters. We predict the TME-specific molecular species that can modulate the ICI response for the ICI-insensitive subtypes.

Further, our model predicts multiple compositional possibilities within the immune-desert subtype due to the different mechanisms behind immune depletion. An exhaustion-driven immune desert leads to the colonization of the immune landscape by exhausted and regulatory T cells. Similar observations have been reported in the recent work by Bao *et al.* (2024), wherein a spatial transcriptomic analysis of the immune module for twenty-six colorectal cancer patients revealed the non-responder to chemotherapy can be attributed to the immune-cold condition, defined as the abundance of regulatory T, and exhausted CD8+ and CD4+T cells [98]. Leveraging the integrative framework, our model predicts that the CAF-promoted immune cold scenario can lead to immune depletion due to the effective shutdown of the helper T cells, and the immune landscape can be hijacked by the regulatory T cells. A high exhaustion rate can also render an immune scenario colonized primarily by the exhausted and regulatory T cells.

Our analysis shows that these identified TME compositions in pre-ICI settings govern the prognosis during and after ICI therapy. Our simulation studies indicate that two pre-ICI TME compositions, resembling 1.) the immune desert and 2) fibro-dominated TME subtypes remain primarily insensitive to ICI therapy. While the ICI resistance due to the immune-desert phenotype is well established, the recent work by Quek *et al.* (2024), from the single cell analysis of five melanoma patients, showed the existence of resistant TME subtypes with abundant CAF and non-zero killer T cell population—This supports our result depicting a fibro-dominated scenario being insensitive to ICI therapy [99]. Lack of clinical response in several HNSCC ICI clinical studies can be mapped to the immune-desert nature of the underlying TME. The post-ICI tumor cell count remains significant despite a meager increase in the proportion of CD8+killers to exhausted T cells [102,103]. According to our model, although an ICI intervention increases the proportion of killers to exhausted T cells, the post-ICI tumor cells remain significant due to the oncogenic role of the exhausted T cells. Our model predicts a reduction of elevated lactate levels may improve the scenario by increasing the effective cytotoxicity of the killer T cells, thereby leading to an aggressive immune response.

Interestingly, according to our model, the abundant CAF population can also lead to a condition wherein the TME, despite a significant killer T cell population in the neighborhood of tumor microenvironment, exhibits an immune-desert scenario vis-a-vis the ICI response. Recent work by Xiao *et al.* (2023) on the effect of CAF on the CAR-T cell therapy for desmoplastic pancreatic tumor patients revealed that the resident CAF population inhibits the killer T cell-driven immune response via the construction of a physical barrier between the T and Tumor cells [103]. The immuno-suppressive role of CAFs and the associated pathways have also been reported in the literature [100–103]. Bao *et al.* (2024) observed a class of non-respondent (to immunotherapy) melanoma tumors with a non-zero killer T cell population [98]. Further, Jiang *et al.* (2019), obtained a strict negative correlation signature between the low tumor infiltrating T cells and the CAF population for 317 metastatic melanoma patients (source: TCGA dataset) [112]. Similarly, Our model predicts that beyond a threshold population levels of CAF, the immune response deteriorates drastically. Further, the CAF-driven barrier divides the local tumor cells into immune-accessible and inaccessible classes. Further, due to resource competition, the application of anti-PD1 increases inaccessible immune cells, rendering the total tumor cell count largely unaffected.

On the other hand, under the parametric conditions resulting in immune/non-fibrotic and immune-dominated TME subtypes, the proposed model yields preferable outcomes to ICI. As suggested by the model, unlike the immune/non-fibrotic subtype, the pre-ICI diagnosis of an immune-dominated TME can potentially present with abundant CAFs and killer T cells. The central differentiating feature between the pre-ICI conditions for fibro-dominated and immune-dominated subtypes is the presence of CAF-protected, immune-inaccessible tumor cells. *In this sense, it may be hypothesized that fibro-dominated is an advanced scenario of immune-dominated TME.*

The model-guided approach enables us to propose possible molecular targets to circumvent the poor prognosis caused by the pre-ICI stable TME compositions (Table 1). Our study shows that non-responsive outcomes to ICI treatment due to an immune-desert TME may be circumvented by a one-time IL-2 spike, which reprograms the TME towards an

immune-hot scenario. Multiple studies on different cancer types, including melanoma, reported that the IL-2-based treatments [104,105], in some scenarios, lead to a complete or partial response [104]. Our model predicts that a one-time IL-2 spike beyond a threshold can drive the immune desert HNSCC TME toward an immune-hot scenario. According to our model, immune-hot TME can only yield a desirable prognosis in immune-accessible tumors, *i.e.,* if the TME composition is not fibro-dominated. This can explain the scenario where an IL-2-based combination therapy does not yield a favorable response. Although the history of IL-2 based treatment dates back to the some of the earliest immune based therapies and carried extensive toxicity, new strategies of delivery, dosing and timing could be considered to shift a TME rather than use to drive the entire therapeutic approach [106].

Unlike the immune-desert TME, the model proposes an OPN + LIF-based intervention for the fibro-dominated scenario. As observed in this study, the OPN and LIF knockout significantly improves the accessibility landscape via a drastic reduction of the CAF population inside the TME. This is also supported by the experimental observations of elevated OPN and LIF levels in ICI-resistant patients in across different cancers [107–111]. Therefore, the two non-responsive TME compositions, namely the fibro-dominated and the immune-desert, require different interventions (apart from the standard ICI) to improve the prognosis. Therefore, it is essential to differentiate these two different TME compositions. Our model proposes that IL-8, unlike most molecular species, may have distinct features for the TME subtypes. In a recent study on the response of HNSCC patients to ICI across HPV signatures, Hill *et al.* (2023) reported that the HPV-negative HNSCC patients presented with increased levels of IL-8 (compared to pre-neoadjuvant ICI setting) in the scenario of no-response. Meanwhile, the post-ICI IL-8 levels significantly reduced the number of responders [93]. Our model predicts that the post-ICI IL-8 remains high and identical to the pre-ICI counterpart in an immune-desert scenario due to the lack of significant change in the tumor cell and M2 macrophage population.

In contrast, the post-ICI IL-8 levels increase for a fibro-dominated scenario compared to its pre-ICI counterpart. This can be reasoned as follows: The post-ICI scenario for fibro-dominated and non-immune desert HNSCC TME can be identified with a significant (marginal) increase in killer T cells, which in turn leads to an increase in the IL-10 levels. Further, elevated IL-10 levels result in aggressive conversion to M2 macrophages from the resident M1 macrophages. Therefore, the overall population of all the IL-8 contributors in the HNSCC TME, the CAF, tumor cells, and M2 macrophages shows a marginal increase compared to its post-ICI counterpart.

Similar to the IL-8 scenario, our model predicts high levels of post-ICI lactate in the fibro-rich scenario. Elimination of lactate, on the other hand, can significantly improve the immune response against the tumor cells. These observations are closely aligned with the existing knowledge on the immune-suppressive role of lactate in cancer [95].

The proposed model primarily serves as an integrated mechanistic framework that maps the existing knowledge about the diverse cell-cell interactions vis-à-vis HNSCC to the clinical outcome in the context of immune checkpoint inhibition and additionally, identifies potential target species and biomarkers towards better prognosis. For this purpose, the model requires the concentration profiles of different cell states mentioned in Fig 2(a) that can be obtained through the proportional measurements of the cell counts (using software such as Cell profiler) over multiple time points. Conversely, the proportion can also be obtained through the dynamic single cell RNA sequencing of the HNSCC primary tumor. Using the t-distributed stochastic neighborhood embedding, it is possible to find out the relative population of each cell state mentioned in the model. Furthermore, the plasma level concentration measurements of the molecular species (chemokines, cytokines, and lactate) are necessary for assessing the veracity of the model predictions.

The proposed model is limited by its scope and granularity and, the objective of this work. The crucial notion of *staging and metastasis* of the primary tumor over time is yet to be a part of this model. This can inspire further modification and complexity over and above the existing model schema. Further, the curious case of distinguishing between human papillomavirus-driven (HPV-positive) and non-HPV based mechanisms of oncogenesis for HNSCC does not figure out in this model. However, it has been well reported that the HPV positive HNSCC yields better response to ICI treatment compared to HNSCC negative scenarios primarily due to an increased immune activation and infiltration program with

the TME for HPV positive patients [115]. Therefore, the model simulations that correspond to a low immune cells or low immune accessibility can be associated with HPV-negative scenarios. Moreover, like any mathematical modeling exercise, the correctness of the conclusions drawn from the proposed model is conditioned on the specific dynamic and the assumptions thereof. Throughout this work, the proposed modeling and analysis framework assumes spatial homogeneity, at least for the molecular species. Therefore, the current model does not factor in the intra-tumoral heterogeneity. This implies that the diffusion current of the molecular species is faster than the timescales of interactions. Additionally, the application of Michaelis-Menten or Hill-like kinetics for the paracrine and autocrine interactions assumes (a) the quasi-steady state of the ligand-receptor complexes and (b) deportation of the product species faster than the reaction timescales which may not be true during the initial stages of the tumor formation. Further, since the model is built from the existing literature, there may exist a degree of confusion about the exactness of the proposed HNSCC TME network. Additionally, although the proposed model attempts to provide a comprehensive picture of the TME, it is not a complete and extensive description of TME, leaving the scope for further modification and updating of the current version. Further, the proposed model has been assessed against the state of the art rules for model development. S6 Text details the self-assessment report.

Overall, the proposed model is able to explain a wide range of clinical observations across patients and contains the scope to be tuned to particular patient-specific quantitative information, including the single-cell transcriptome analysis and immunohistochemistry of the TME. Therefore, the proposed model albeit, not meant for direct clinical deployment, can serve as a crucial first step for the development of personalized (clinically deployable) model. The next step requires leveraging the patient-specific dynamic single cell transcriptomic analysis and immune-histochemistry towards construction of population-specific distribution of key parameters in human cohorts.

## Materials and methods

This section lays out the details of model construction.

### Reconstruction of the TME components and interactions from literature

We first reconstructed the TME network structure for HNSCC from the literature. Subsequently, we used rate laws applicable to each of the interaction types to formulate a quantitative systems model of HNSCC TME. Each edge in the proposed network (Fig 2(a) of the revised manuscript) is supported by at least one experimental evidence published in the literature of HNSCC or similar solid tumors such as melanoma.

According to the standard practice of the mathematical modeling, the proposed model should capture the necessary interactions sufficient to meet the model objectives in a way that does not result in an increased complexity and downstream identifiability issues. Therefore, the present version of the model does not include a number of additional cellular states that have been shown to play an important role in the context of HNSCC but can be functionally represented by a different cell type. For instance, the existence of B cells and the role of different cell states with B cells have been widely explored in the context of HNSCC [34]. The mature B cells transform into an activated state in the presence of an antigen via the helper T cells. Further, the activated B cells produce antibodies which, upon binding with the antigen presenting tumor cells can facilitate the T cell-driven killing of the tumor cells. Based on this understanding, the destruction of tumor cells by B cells are functionally captured by incorporating the tumor cell-dependent growth of helper T cells and helper T cell stimulated infiltration of killer T cells. Further, the tumor-promoting role of the exhausted B cells have also functionally been taken into account through the role of the exhausted T cells. Therefore, although the B cells are biology different than the different cell states, the effect of the B cells on the tumor cell count can be approximated by finetuning the parameters (proliferation rate, tumor cell-driven activation rate) pertaining to the helper T cells and cytotoxic T cells.

We include four cell types as the constituting modules of the TME, namely i) tumor cells, 2) T cells, 3) macrophages, and 4) fibroblasts [4,10,36]. We also consider eight molecular species secreted from one or many of the four cell types. Interleukin-2 (IL-2), interleukin-6 (IL-6), interleukin-8 (IL-8), interferon gamma (IFNγ), lactate (Lac), intercellular adhesion

molecule (ICAM), Osteopontin (OPN), and interferon regulatory factor-8 (IRF-8)— these eight molecular species are crucial for the conversion, proliferation, and apoptosis of the cell types present in the TME.

### Tumor cells

Characteristic to the TME, tumor cells are associated with high epithelial scores and high mutational burden [34]. Recent single-cell transcriptome analysis revealed the diversity within the tumor cells. In the scenario of HNSCC, Puram *et al.* (2017) proposed seven distinct tumor cell states with diverse groups of biomarkers [34]. The transcriptome analysis of a number of cancer cell lines reveals a specific stem-like pattern which, upon further analysis has been reported to transition into different tumor cell states to facilitate immune escape, metastasis, and drug resistance [4,6,19,30,32]— this may also be understood as the phenomena of 'cooperation' between tumor cells [38].

Since this work is centered around immuno-therapy, we classify the tumor cells based on their response to the activities of the immune cells. It is well-known that the natural killer (NK)-cells and the killer T cells recognize the tumor cells without programmed death ligand1 (PDL1) [23,39]. Interestingly, several studies confirmed that a PD-L1 expression in tumor cells is a known strategy for immune escape for solid tumors [10,23,41].

Therefore, the proposed formalism consists of three different tumor cell states— 1) tumor stem cells, 2) tumor cells without PDL1, and 3) tumor cells with PDL1 expression. Further, the tumor cells secrete inflammatory cytokines (such as OPN, IL–8) and lactate that aid in self–proliferation and sustenance in the presence of immune onslaught [42–45].

### T-cells

The T-cells, as the main driver of the immune response, destroy the stem-like and PDL1−tumor cells. The killer and helper T-cells also secrete many pro-immune cytokines such as IFNγ, IL-2, and ICAM1 [43–45]. Upon a classification exercise of the transcriptome of the non-tumor cells, Puram et al. (2017), identified four different cell states within the T-cell cluster— namely 1) PD1+killer, 2) helper T cells, 3) regulatory T cells, and 4) exhausted T cells [34].

The helper T cells promote the growth of the killer T cells in the presence of PDL1−tumor cells [46]. On the other hand, regulatory T-cells control the population of helper T cells to circumvent the overactive immune response [36]. Further, there also exists a significant population of killer-like T cells (also known as exhausted T-cells) that are derived from the interactions between the PD1+killer T cells, PDL1+tumor cells, and M2-phase macrophages. With ICI-based therapies, the killer PD1+T cells are converted to PD1−, thereby reducing the conversion flux towards the exhausted T cells and improving the overall immune response [35,51].

### Macrophages

Macrophages play a dual role within the TME. The M1 phase macrophages aid in the production of killer T cells [23,52]. Further, the M1 macrophages also secrete transcription factors that can play a crucial role in inhibiting the secretion of pro-inflammatory cytokines. The M1 macrophages also secrete IRF8, inhibiting OPN secretion in TME, thereby improving the prospect of ICI therapy.

Unlike the M1 phase, the M2 phase macrophages serve a pro-tumor role in several ways, such as accelerating the conversion of killer T cells to an exhausted state [53,54].

### Fibroblasts

Fibroblasts are the most common unit apart from the tumor cells to be found inside the TME. Broadly classified in two distinct phenotypes– myofibroblasts and invasive, fibroblasts modulate the proliferation flux of the tumor cells via paracrine interactions. Further, the invasive fibroblasts can also protect the tumor cells from being attacked by the killer T cells [55,56]. Additionally, fibroblasts also secrete several cytokines (IL-8 and OPN) that can further the growth of tumor cells and hamper the immune response via the CAF-mediated proliferation of regulatory T cells [57,58,70].

## Molecular species

The present modelling approach considers several cytokines, chemokines and metabolites that are secreted and produced by different cell types and in turn, mediate intercellular interactions. For instance, as reported by Liu et al. [81], OPN plays an important role in the proliferation of tumor cells through modulating the MAPK pathway [81]. Similar observations are also reported for gastric cancer and bone cancer [82]. The IL-6 family cytokine LIF aids in the conversion of the wild type fibroblasts into a more invasive tumor-facilitating CAF phenotype [69]. On the other hand, the role of IL-2 in promoting the proliferation of killer T cells has been well documented in the solid tumor research [104–106]. Interestingly, the dual role of IFNγ in sensing the presence of tumor cells (thereby activation of helper T cells) and inducing PDL1 expression in tumor cells leading to immune escape of tumor cells have been reported for a wide range of solid tumors including HNSCC [77–79]. Although the exact role of IL-8 in the context of HNSCC remains elusive, several experimental studies report a strong correlation between the IL-8 expression and metastasis, and ICI resistance [93]. Therefore, we place IL8 as the main factor controlling the transitions from the stem-like tumor cells to PDL1- and +ve tumor cells.

## Modeling of the TME dynamics

This section presents the specific interactions and crosstalk between different constitutive units discussed in the preceding sections. We establish the general modeling rules and assumptions adopted throughout this work.

A.1.    **Rules.** The following are the general rules adopted for the model.

1. The dynamics is modeled using ordinary differential equations (ODE) i.e., the cell-state populations and the molecular concentrations are modeled as functions of time.

2. Proliferation: Each cell state ($x_j$) has its own proliferation flux expressed in terms of logistic growth [24]

$$f_j^{prol} = K_j^{prol} x_j \left( 1 - \frac{x_j}{x_j^{max}} \right)$$

where, $K_j^{prol}$, and $x_j^{max}$ are the associated rate for proliferation and the carrying capacity of $x_j$ respectively.

3. Conversion: The conversion from a cell state ($x_j$) can to a different cell state ($x_i$) within the same cell type is modeled using first order kinetics [30,31] with the proposed flux rate

$$f_{j,i}^{conv} = K_{j,i}^{conv} x_j$$

where, $K_{j,i}^{conv}$ is the associated conversion rate.

4. Paracrine interaction: If a cell-state $x_i$ promotes the proliferation rate of cell xj then the modified proliferation rate for $x_j$ is expressed using a rational (Hill-kinetics like) dynamics [113]

$$\widetilde{f_j^{prol}} = f_j^{prol} \left( 1 + K_{j,i}^{para} \frac{\alpha_{j,i} x_i}{V_{i,j}^M + a_{i,j} x_i} \right)$$

where, $K_{j,i}^{para}$, $V_{j,i}^M$, and $\alpha_{j,i}$ are the rate constant, dissociation constant, and the spatial proximity index between $x_i$ and $x_j$ respectively.

5. Regulatory inhibition: The regulatory inhibition between two species $x_i$ (source) and $x_j$ (target) modulates the proliferation in the manner proposed by Sontag and coworkers (2016) in the work of constructing toggle switches via regulatory inhibition [114]

$$f_{j,i}^{reg} = \left( \frac{1}{V_{j,i}^{reg} + \alpha_{j,i} x_i} \right)$$

where, $V_{j,i}^{reg}$, is the dissociation constant for the regulatory inhibition.

6. Elimination: If a cell state $x_i$ eliminates/ destroys another cell state $x_j$ the associated flux can be expressed as [24]

$$f_{j,i}^{kill} = K_{j,i}^{kill} \alpha_{j,i}^{ac} x_j x_i$$

where, $K_{j,i}^{kill}$ is the rate constant corresponding to the elimination reaction. The term $\alpha_{j,i}^{ac}$ modulates the accessibility (proximity) from the cell state $x_i$ to $x_j$.

7. Death/Degradation: Every species is assumed to have a natural death (for cell states) or degradation (molecular species) rate that is reflected in the following manner [24]

$$f_j^D = K_j^D x_j$$

where, $K_j^D$ is the natural death/ degradation rate.

**A.2. Calculating the proliferation to death balance ($\zeta$)**

As illustrated from our study, the clinically observed TME subtypes can be mapped as the pre-ICI steady state of the proposed TME model for HNSCC. For this purpose, we introduced a quantity $\zeta$ that is defined as the ratio between the maximum proliferation factor of a given cell state to the corresponding death and conversion rate. We calculate the maximum proliferation factor($F_x^{max}$) for a given cell state (x) in the following manner:

$$(F_x^{max}) = K_x^{prol} \prod_{j \in S_x} \left( 1 + K_{\{x,j\}}^{para} \right)$$

Where, $S_x$ is the set of all the incoming, neighbor nodes of x. It is to be noted that since the paracrine interaction dynamics, according to the proposed model, is rational, polynomial bounded above by the paracrine rate constant the net proliferation rate of a cell state is always bounded above by ($F_x^{max}$).

On the other hand, the depreciation of the cell state x occurs through the processes of killing, conversion, and natural death. Therefore, the log-transformed net proliferation to death balance $\zeta$ can be calculated as

$$\zeta = \log \left( \frac{F_x^{max}}{K_{x,j}^{kill} + K_{x,j}^{conv} + K_x^D} \right)$$

Now, for total tumor cells, we used the maximum $\zeta$ across different tumor cell states.

**CAF-mediated alteration of immune accessibility**

It is well-known in literature that CAF plays a crucial role in remodeling the extra cellular matrix and thereby altering the accessibility from the T cells to the tumor cells. To address this critical phenomenon in the proposed model, we defined the quantity immune accessibility index ($0 \leq I_a \leq 1$) - a bounded hyper-parameter defined as the ratio between the total

carrying capacity of immune accessible tumor cells to that of fibroblast protected tumor cells. We also proposed an expression for the immune accessibility index. We observed that the lack of immune accessibility alters the TME towards a fibro-dominated environment. We propose the following way to calculate immune accessibility index ($I_a$)

$$Ia := \frac{K_{IA}}{K_{FP}} := 1 - \tanh\left(\alpha . CAF . K_{br}\right)$$

where, $K_{IA}$ and $K_{FP}$ denote the carrying capacities of immune-accessible and CAF-protected tumor cells respectively, $\alpha$ determines the fraction of CAFs near the tumor cells, and $K_{br}$ is the barrier formation rate for the CAFs. As depicted in the expression for the immune accessibility index, is a bounded quantity in the closed interval [0,1] wherein $I_a = 1$ refers to the situation where all the tumor cells are accessible to the immune system, whereas $I_a = 0$ denotes the impenetrability of the immune cells to the TME.

As established before, the CAF population around the tumor cell states significantly affects immune accessibility. Therefore, for any given CAF population, $\alpha$, and $K_{br}$ we propose the following rule for the penetration of the immune cells ($\widetilde{F_{CD}}$) to CAF protected tumor cells as

$$\widetilde{F_{CD}} = F_{CD}e^{-\alpha^2 \delta CAF^2}$$

where $\delta$ is the width of CAF barrier.

## Simulation

All the simulations were performed using MATLAB 2022B (Mathworks Inc., Natick, MA). The ordinary differential equations in the models are simulated using ode23s in MATLAB 2022B and have also been reproduced in Python Jupyter Notebook with the help of 'ivp' solver and BDF method. All the MATLAB and Pyhton notebook codes required for replication and reproduction of the results are publicly available on the Github repository.

## Supporting information

**S1 Text. Mathematical representation of the fluxes corresponding to the HNSCC TME network in S1 Fig below.** (PDF)

**S2 Text. Overall mathematical model.** (PDF)

**S3 Text. Parameters varied for Figure 2.** (PDF)

**S4 Text. Parameters fixed for Figure 2.** (PDF)

**S5 Text. Abbreviations.** (PDF)

**S6 Text. Self-assessment of the model using ten simple rules.** (PDF)

**S1 Fig. Detailed HNSCC TME network model. (a)** The nodes are either the cell states or the molecular species, whereas the edges represent diverse forms of interactions. The acronyms C_0, C +, and C− refer to stem, PDL1+

(programed death ligand1), and PDL1– tumor cells, respectively. T_K+, T_K-, T_Help, T_Reg, and T_EX stands for PD1+ (programmed death 1), PD1- killer T cells, Helper T cells, Regulatory T cells, and Exhausted T cells, respectively. M_1 and M_2 refer to macrophages of M1 and M2 phase, respectively. Further, F_WT and CAF correspond to wild type and invasive cancer associated fibroblasts, respectively. The acronyms IL-2, IL-8, IL-10 LIF, IFNG, IRF8, OPN, ICAM1, and Lac denote Interleukin 2, Interleukin 8, Interleukin 10, Leukemia Inhibitory Factor, Interferon Gamma, Interferon Regulatory Factor 8, Osteopontin, Intercellular Adhesion Molecule 1, and Lactate, respectively. All the cell states are assumed to be capable of self-proliferation and natural death. Therefore, the self-loops are not shown for brevity. **(b-d)** Each tumor cell state is subdivided depending on the accessibility from the Killer T cells. The Killer T cell-exposed tumor cells are exposed to immune response whereas the Killer T cell-non-exposed tumor cells are protected by the CAF-derived barrier from immune onslaught. **(e)** Flux-structure mapping for the killer T-cell-expose tumor stem cells.
(TIF)

**S2 Fig. Killer T cell-independent growth of CAF.** The proliferation rate governs the pre-ICI population of killer T cells. Below a critical proliferation rate the HNSCC TME model settles in an immune-desert region. Whereas, in both the scenarios (immune-desert and immune-rich), the CAF population remains unaltered indicating a relative independence from the T cell population.
(TIF)

**S3 Fig. Resource intake governs overall dependence of helper T cell on pro-tumor role of CAF. (a-d)** The pro-tumor role of CAF leads to significant pre-ICI, PDL1- tumor cell population. Therefore, beyond a threshold value of the pro-tumor role of CAF (compared to the CAF-driven growth of regulatory T cells), the final helper T cell population remains high. On the other hand, for moderate to low CAF-tumor interaction, the PDL1- tumor cells remain low due to the presence of cytotoxic killer T cells. Therefore, despite an initial increase, the helper T cells settle to a very low value (almost zero). Further, the threshold value of CAF-tumor interaction is dependent on the maximum resource intake in a competitive setting.
(TIF)

**S4 Fig. ICI reduces CAF population and increases the wild type fibroblasts in immune rich scenario. (a)** The CAF population exhibits a steep increasing tendency owing to multiple paracrine interaction with the tumor cells and tumor associated macrophages. However, the ICI intervention in an immune rich scenario reduces the tumor cells. Further, the reduction in tumor cells-secreted LIF reduces the transition flux from wild type to cancer-associated fibroblasts. Therefore, overall CAF population undergoes a significant reduction during the ICI therapy. **(b)** The wild-type fibroblast population, due to significant reduction in the transition flux towards CAF, increases during an ICI-based therapy in immune rich scenario.
(TIF)

**S5 Fig. Exhaustive evaluation of ICI response for immune-dominated scenario. (a)** Possible tumor cell, Killer T cell, and CAF populations within the subset of the parameters (selected for Figure 2) corresponding to the immune-dominated subtype (Group-4). The application of anti-PD1 results in the significant reduction (complete removal in a few scenarios) of the tumor cells, CAF population. On the other hand, the anti-PD1significantly increases the resident killer T population in the TME.
(TIF)

**S6 Fig. The CAF-immune accessibility story. (a)** Demonstrates the phase-space between the immune accessibility and CAF for different CAF proliferation rate. **(b)** The time profile for immune accessibility shows the existence of a

threshold time beyond which the immune accessibility deteriorates drastically. Further, this threshold time is dependent on the proliferation rate of CAF. This is due to the fact that a higher proliferation rate renders a faster CAF growth and due to the near-linear trajectory of CAF-immune accessibility trajectory, the immune accessibility adopts a faster time scale.
(TIF)

**S7 Fig. IL2-Killer T cell story. (a-d)** Increasing levels of IL-2-induced killer T cell proliferation rate can drive the HNSCC immune-desert TME to an immune-hot scenario. However, there exists a threshold IL-2-driven Killer T cell proliferation rate below which the immune-desert scenario can not be circumvented irrespective of the external IL-2 level.
(TIF)

**S8 Fig. LIF knockout reduces increases immune accessibility. (a)** Phase trajectory of different tumor cells vis-à-vis immune accessibility. The LIF knockout significantly reduces the pre-ICI inaccessible tumor cells. However, a complete (or near) complete elimination of inaccessible tumor cells is not possible with only LIF knockout. **(b)** Although a LIF knockout improves the immune accessibility it does not drive the TME system to an immune-dominated situation.
(TIF)

**S9 Fig Sensitivity analysis.** We chose all the parameters that exhibits an explicit bearing with the proliferation and death and conversion fluxes for Tumor cells (Blue), Killer T cells (Violet), and CAF (Yellow).
(TIF)

## Acknowledgments

The authors thank Alexandra Machel for reviewing the MATLAB code and replicating the figures of the main manuscript. The authors are thankful to Prem Jagadeesan for independently implementing the model in MATLAB based on the manuscript text and supplement to reproduce the Figure 3 of the main manuscript.

## Author contributions

**Conceptualization:** Priyan Bhattacharya, Alban Linnenbach, Larry A. Harshyne, Mỹ G. Mahoney, Adam J. Luginbuhl, Rajanikanth Vadigepalli.

**Data curation:** Alban Linnenbach, Andrew P. South, Ubaldo Martinez-Outschoorn, Mỹ G. Mahoney, Adam J. Luginbuhl.

**Formal analysis:** Priyan Bhattacharya.

**Funding acquisition:** Alban Linnenbach, Andrew P. South, Ubaldo Martinez-Outschoorn, Joseph M. Curry, Jennifer M. Johnson, Larry A. Harshyne, Mỹ G. Mahoney, Adam J. Luginbuhl, Rajanikanth Vadigepalli.

**Investigation:** Priyan Bhattacharya, Rajanikanth Vadigepalli.

**Methodology:** Priyan Bhattacharya, Adam J. Luginbuhl, Rajanikanth Vadigepalli.

**Project administration:** Adam J. Luginbuhl.

**Resources:** Adam J. Luginbuhl, Rajanikanth Vadigepalli.

**Supervision:** Rajanikanth Vadigepalli.

**Writing – original draft:** Priyan Bhattacharya.

**Writing – review & editing:** Priyan Bhattacharya, Alban Linnenbach, Andrew P. South, Ubaldo Martinez-Outschoorn, Joseph M. Curry, Jennifer M. Johnson, Larry A. Harshyne, Mỹ G. Mahoney, Adam J. Luginbuhl, Rajanikanth Vadigepalli.

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
