## [Decision Letter · Decision Letter 0]

PCOMPBIOL-D-25-00098

Tumor microenvironment governs the prognostic landscape of immunotherapy for head and neck squamous cell carcinoma: A computational model-guided analysis

PLOS Computational Biology

Dear Dr. Vadigepalli,

Thank you for submitting your manuscript to PLOS Computational Biology. After careful consideration, we feel that it has merit but does not fully meet PLOS Computational Biology's publication criteria as it currently stands. Therefore, we invite you to submit a revised version of the manuscript that addresses the points raised during the review process.

Please submit your revised manuscript within 60 days May 01 2025 11:59PM. If you will need more time than this to complete your revisions, please reply to this message or contact the journal office at ploscompbiol@plos.org. Please include the following items when submitting your revised manuscript:

We look forward to receiving your revised manuscript.

Kind regards,

Prakriti Mudvari

Academic Editor

PLOS Computational Biology

Mark Alber

Section Editor

PLOS Computational Biology

**Journal Requirements:**

At this stage, the following Authors/Authors require contributions: Priyan Bhattacharya, Alban Linnenbach, Andrew P. South, Ubaldo Martinez-Outschoorn, Joseph M. Curry, Jennifer M. Johnson, Larry A. Harshyne, Mỹ G. Mahoney, Adam J. Luginbuhl, and Rajanikanth Vadigepalli. Please ensure that the full contributions of each author are acknowledged in the "Add/Edit/Remove Authors" section of our submission form.

2) We note that your Supplementary Figures files are duplicated on your submission. Please remove any unnecessary files.

5) We notice that your supplementary figures are uploaded with the file type 'Figure'. Please amend the file type to 'Supporting Information'. Please ensure that each Supporting Information file has a legend listed in the manuscript after the references list.

Potential Copyright Issues:

i) Thank you for stating that "The figures have been created using Biorender.com." Please confirm that you hold a Premium account and provide a pdf copy of the CC BY 4.0 Licence as provided by BioRender. For instructions on how to generate a CC BY 4.0 license for your figure, please see the guidelines here: https://help.biorender.com/hc/en-gb/articles/21282341238045-Publishing-in-open-access-resources.

If you are using the free assets from BioRender, we are unable to publish these images as they are licenced under a stricter licence than CC BY 4.0. In this case we ask you to remove the BioRender images and replace them with open source alternatives.

See these open source resources you may use to replace images / clip-art:

- https://openclipart.org/

7) Please amend your detailed Financial Disclosure statement. This is published with the article. It must therefore be completed in full sentences and contain the exact wording you wish to be published.

2) If the funders had no role in your study, please state: "The funders had no role in study design, data collection and analysis, decision to publish, or preparation of the manuscript."

8) Please ensure that the funders and grant numbers match between the Financial Disclosure field and the Funding Information tab in your submission form. Note that the funders must be provided in the same order in both places as well. Currently, the order of the funders is different in both places.

**Reviewers' Comments:**

Reviewer #1: The biology examined in the paper is not my area of expertise so I won't comment on the quality of the biological research. Instead I will make some, hopefully useful comments, on the modeling approach. The model is a straight forward, though non-trivial reaction/interaction based model, that can be described using differential equations. The authors chose to use MATLAB to distribute the model. What this means is that the model cannot be easily reused and requires a third-party to purchase a proprietary license for MATLAB which reduces access. The use pf Python or Julia would address access but not the reuse issue. In addition, there is now ample evidence (Bayesian estimation reveals that reproducible models in Systems Biology get more citations Sebastian Höpfl et al), that shows that models encoded in open standards are more cited and therefore have more impact that ones that are not. The authors do provide the MATLAB code which is quite dense on inspection so in principle the work can be repeated but no easily reproduced. It also means the model won't appear in any of the major model repositories, which is unfortunate. I am not expecting the authors to recode their model other than to suggest in the future they might use more open science based methodologies to publish their work. I also note their use the 10 rules of credible practice but only appear to conform to 3 of the rules. Admittedly this is better than most :)

Reviewer #2: In this manuscript, Bhattacharya et al model key cellular and molecular interactions within a tumor microenvironment to understand resistance to immune checkpoint inhibition (ICI) therapy. Overall, this is a comprehensive piece of work focused on profiling the composition of the tumor microenvironment. It also maps the response to different immune therapies based on cellular composition. The models are generated based on specific hypotheses and a breadth of scenarios are discussed in the manuscript. Overall, this is a strong and well-executed study. However, I have a few concerns:

1. The overall narrative is somewhat fragmented. The manuscript could benefit from some consolidation and a better-focused storyline.

2. The network in Fig. 2 is not described in sufficient detail in the text. A more comprehensive description of Fig. 2A should be added.

3. It is difficult to tell the differences between Figs. 3e and 3f (with and without lactate). These differences should also be appropriated linked to corresponding conclusions.

4. The authors do a good job of describing their model assumptions. However, they should provide additional details regarding specific biological contexts where these assumptions hold vs where they don't.

5. There are some typos and incorrect figure calls. These should be corrected.

Reviewer #3: Bhattacharya et al describe in their manuscript a model that governs the response to immunotherapy. The mathematical modeling part is impressive, but I believe that the current version of the manuscript lacks three major parts: 1. Detailed description of how the selection of parameters was done. 2. Description of how this model would be used in the clinic. 3. Any kind of validation that the proposed model actually works. Adding these three items would significantly improve the manuscript and its impact.

1. Detailed description of how the selection of parameters was done. The current version lacks any kind of detailed description of why certain parameters were selected and, more importantly, why some obvious parameters are missing. For example, it is well established that B cells play an important role in immune response and this is not included. I’m also surprised that HPV status is not considered or mentioned. This may be all correct, but it does warrant some discussion. Would a non-relevant parameter still generate similar results as the selected ones? Were other parameters initially tested but did not work that well and were thus left out in the final version? Should ease of measurement be considered when selecting parameters?

2. Description of how this model would be used in the clinic. From a practical point of view, how would the model translate to a CLIA-approved test? What measurements are needed to derive the required input variables for a patient? How would previous treatments, such as radiation and/or chemo change the results? Does the model require spatial transcriptomics data, or would some much more simple measurements work? This is important to know to be able to understand the clinical relevance of the manuscript.

3. Any kind of validation that the proposed model actually works. In the current version, we have no idea if the proposed model actually works. This makes all the discussion and questions asked in the Result section questionable. Basically, Figure 2 to Figure 10 all assume that the model is correct. It feels a bit redundant that so much space is given to interpret a model that the authors built themselves. Could any public dataset be used to show that the model works. There must be some spatial transcriptomics or single-cell dataset available.

**Have the authors made all data and (if applicable) computational code underlying the findings in their manuscript fully available?**

Reviewer #1: Yes

Reviewer #2: Yes

Reviewer #3: Yes

PLOS authors have the option to publish the peer review history of their article (what does this mean? ). If published, this will include your full peer review and any attached files.

**Do you want your identity to be public for this peer review?** For information about this choice, including consent withdrawal, please see our Privacy Policy .

Reviewer #1: No

Reviewer #2: No

Reviewer #3: No

**Figure resubmission:**
---

## [Decision Letter · Decision Letter 1]

Dear Dr. Vadigepalli,

We are pleased to inform you that your manuscript 'Tumor microenvironment governs the prognostic landscape of immunotherapy for head and neck squamous cell carcinoma: A computational model-guided analysis' has been provisionally accepted for publication in PLOS Computational Biology.

Best regards,

Prakriti Mudvari

Academic Editor

PLOS Computational Biology

Mark Alber

Section Editor

PLOS Computational Biology

Reviewer's Responses to Questions

**Comments to the Authors:**

Reviewer #2: The authors have suitably addressed all my comments. The revised manuscript overall represents an interesting piece of work.

Reviewer #3: They have tried to address my concerns and comments.

**Have the authors made all data and (if applicable) computational code underlying the findings in their manuscript fully available?**

Reviewer #2: Yes

Reviewer #3: Yes

PLOS authors have the option to publish the peer review history of their article (what does this mean? ). If published, this will include your full peer review and any attached files.

**Do you want your identity to be public for this peer review?** For information about this choice, including consent withdrawal, please see our Privacy Policy .

Reviewer #2: **Yes: ** Jishnu Das

Reviewer #3: No

---

## [Editor Report · Acceptance letter]

PCOMPBIOL-D-25-00098R1

Tumor microenvironment governs the prognostic landscape of immunotherapy for head and neck squamous cell carcinoma: A computational model-guided analysis

Dear Dr Vadigepalli,

I am pleased to inform you that your manuscript has been formally accepted for publication in PLOS Computational Biology. Your manuscript is now with our production department and you will be notified of the publication date in due course.

With kind regards,

Zsofia Freund
